# Critical Assessment of Membrane Technology Integration in a Coal-Fired Power Plant

**DOI:** 10.3390/membranes12090904

**Published:** 2022-09-19

**Authors:** Maytham Alabid, Calin-Cristian Cormos, Cristian Dinca

**Affiliations:** 1Faculty of Energy, University Politehnica of Bucharest, Splaiul Independenței, 060042 Bucharest, Romania; 2Chemical Engineering Department, Faculty of Chemistry and Chemical Engineering, Babes—Bolyai University, 11 Arany Janos, 400028 Cluj-Napoca, Romania; 3Academy of Romanian Scientists, Ilfov 3, 050044 Bucharest, Romania

**Keywords:** membrane technologies, CO_2_ capture, post-combustion processes, economical assessment, process integration

## Abstract

Despite the many technologies for CO_2_ capture (e.g., chemical or physical absorption or adsorption), researchers are looking to develop other technologies that can reduce CAPEX and OPEX costs as well as the energy requirements associated with their integration into thermal power plants. The aim of this paper was to analyze the technical and economic integration of spiral wound membranes in a coal-fired power plant with an installed capacity of 330 MW (the case of the Rovinari power plant—in Romania). The study modeled energy processes using CHEMCAD version 8.1 software and polymer membranes developed in the CO_2_ Hybrid research project. Thus, different configurations such as a single membrane step with and without the use of a vacuum pump and two membrane steps placed in series were analyzed. In all cases, a compressor placed before the membrane system was considered. The use of two serialized stages allows for both high efficiency (minimum 90%) and CO_2_ purity of a minimum of 95%. However, the overall plant efficiency decreased from 45.78 to 23.96% and the LCOE increased from 75.6 to 170 €/kWh. The energy consumption required to capture 1 kg of CO_2_ is 2.46 MJ_el_ and 4.52 MJ_th_.

## 1. Introduction

In the past decade, considerable improvements in polymeric membrane materials for gas separation processes have been observed, where many types of polymer materials have been manufactured. Consequently, their transport properties could provide an energy-efficient path for wide-range gas separations. Due to the urgent demand to mitigate CO_2_ emissions from the application of fossil fuels, these planned research potentials have been vastly catalyzed. In general, CCS technologies are believed to be a successful technological solution to reduce the environmental and economic carbon dioxide influences, comprising climate change [1]. The classical process for CO_2_ capture, which is based on amine absorption, demands high energy, needs high capital and operating cost, and causes corrosion and environmental troubles in addition to some operational problems. As a result, the membrane is used due to high energy efficiency, operational plainness, system consolidation, and capability to beat thermodynamic solubility restrictions. As a rule, a membrane is a thin interphase layer that plays a significant role in separating two phases [2,3].

Presently, various CO_2_ capture processes have been used based on either ejecting CO_2_ from flue gas (post-combustion), from syngas (pre-combustion), or injecting pure O_2_ instead of air (oxy-fuel combustion) which provides a high CO_2_ flow [4,5,6,7].

So far, different membranes have been enhanced and characterized by high permeability and selectivity to achieve a high CO_2_ capture process. Various types of membrane materials such as inorganic membranes, common polymers, carbon molecular sieve membranes, mixed matrix membranes, fixed site-carrier (FSC) membranes, and carbon molecular sieve membranes have been utilized for CO_2_ separation operation [8]. Nevertheless, to provide a commercially viable membrane to capture CO_2_ and contend with the classical amine absorption process, membrane technology must have a relatively low power consumption and low particular capture cost conjointly with an acceptable stability exposure to impurities such as SO_2_ and NO_x_, which are generally included in the flue gas stream [8].

Over the last decade, preferable polymeric materials with enhanced CO_2_ permeability and selectivity have been vastly researched. Generally, more permeable polymers lead to less selectivity and vice versa, similar to a trade-off relationship. In any case, most polymeric materials depend on the solution-diffusion mechanism in which this trade-off is heavily rooted [9,10]. 

Processability is a basic demand for an effective industrial gas separation membrane. Dense polymers are suitable membrane materials whereas most commercial membranes are manufactured from polymers with low-cost and good scalability, as a result of the credible fabrication of a fine selective layer of 0.1–10 μm and membrane surface area at a range of (1000–1,000,000 m^2^) [11]. The simplest model utilized to demonstrate and predict the gas permeation process via the dense membrane is the solution-diffusion model, where a molecule can transport from one side of the material to another by a concentration gradient. In typical, the separation can be obtained by the difference in solubility and/or diffusivity. The solubility of specific gases relies on their condensability and affinity to the membrane material. Generally, the molecule condensability expands by increasing critical temperature [12]. On the other hand, the diffusion of a gas molecule depends on the space between two chains, where the chain piece’s random motion permits small kinetic diameter molecules to diffuse through [13]. As a result, the molecule diffusivity increases by the decrease of kinetic diameter. Table 1 demonstrates the physical properties of different gases.

## 2. Solution-Diffusion Polymeric Membranes

In this part, the current advances for polymers that are eligible for selective CO_2_ separation based on the mechanism of solution diffusion are researched. Five types of polymers are involved: polyethylene oxide (PEO), perfluoro polymers, polymers of intrinsic microporosity (PIMs), thermally rearranged (TR), and iptycene-containing polymers. These types are sorted depending on the discovery period or application in the gas separation process. Polymer string hardness is the main factor that affects this sequence, from the rubbery PEO to the vitreous TR polymers. In addition, the iptycene-containing polymers are derived/adjusted polymers through the integration of iptycene-containing moieties [15].

### 2.1. PEO-Based Membranes

Despite the CO_2_ molecule being non-polar, the different distribution of charges inside gives the molecule a quadrupole moment [16]. In poly (ethylene oxide) (PEO), the polar ether connection (–C–C–O–) is noticed to have a high affinity to CO_2_ [17]. Thus, PEO-based polymers offer a major CO_2_ solubility, and CO_2_ selectivity typically derives from the solubility selectivity. However, one drawback that has been observed is the high degree of crystallinity in pure PEO or in materials related to PEO. Due to the polar ether groups inclination to compose powerful hydrogen bonding, which leads to a compact chain packing [18]. The crystalline area that formed obstructs the diffusion of CO_2_ and ultimately restrains the membrane permeability. Weak mechanical strength is also a result of high crystallinity. To compensate for these restrictions, different processes have been applied, including (1) block copolymerization with other rigid pieces, (2) combining with low MW poly (ethylene glycol) (PEG) and its derivatives, and (3) crosslinking to compose a highly branched PEO polymer network [19]. In addition to providing noticeably permeable PEO membrane materials, these efforts also drive a remarkable understanding of the nanostructures of the PEO polymers. Generally, the most accepted process to prevent the high crystallinity of PEO is to block the copolymerization of PEO with rigid pieces [11,20]. 

Table 2 demonstrates the permeabilities of CO_2_ and CO_2_/gas selectivities of the PEO polymers based on various strategies.

The PEO block copolymers commonly demonstrated a CO_2_ permeability around 100–200 Barrers with the selectivity of CO_2_/N_2_ about 50 at 25 °C. Monodisperse tetra-amide (T6T6T) and pentiptycene-based polyimide (pent-PI) are kinds of many new hard segments that were incorporated with PEO to assemble ultra-permeable PEO-based copolymers [22,23]. The self-synthesis trait of the PEO-based block copolymer was also largely studied. A polymer chain rearrangement was noticed by Yave et al. when an ultrathin selective layer was covered onto a hydrophobic PDMS face, resulting in a high CO_2_ permeance of 1815 GPU with a CO_2_/N_2_ selectivity around 50 at 30 °C [23]. Xue et al. assembled a PEO–polystyrene (PS) block copolymer, where cylindrical PEO domains are formed, which provide intense CO_2_ permeance of 20,400 GPU with a CO_2_/N_2_ selectivity around 27.7 at 70 °C [24]. 

Another strategy that is utilized to increase the ether content in the polymer matrix is known as blending. A 100–2 000 MW short-chain PEG was established in a PEO-based copolymer via polymer chain tangle, which improved the CO_2_ solubility and broke the compact packing of the PEO piece in the copolymer. The strong hydrogen bonding among the ether kinds can be reduced due to the tip parts on PEG that supplied another control. As a result, PEG moieties with end parts, such as methyl ether, ally ether, divinyl ether, and butyl ether, were researched [23].

The crosslinking process indicates the bottom-up assembly of extreme branches of PEO or PEO-based copolymers by the polymerization of ethylene oxide monomer or oligomer. In this process, the primary work was based on different methacrylate monomers, where the crosslinking mitigated the crystallinity and enhanced the film-forming capacity [28]. Kline et al. demonstrated that the crosslinking density and heterogeneity could be adjusted via the platform of poly (ethylene glycol) diglycidyl ether and polyether diamine, where the heterogeneity crosslinking enhanced the permeability of CO_2_ [27]. In Figure 1, unimodal, bimodal, and clustered PEO networks are assembled.

PEO-based polymers have the main commercial potential, due to the high CO_2_ permeance that has been presented in thin-film-composite membranes under specific testing conditions. Furthermore, the various PEO-based materials that have demonstrated practicable selectivities at an operating temperature of more than 35 °C are of special significance [24,29]. 

### 2.2. Perfluoro-Polymers

These are a group of glassy hydrocarbon polymers with added fluorine atoms instead of all or most hydrogen atoms. Due to the powerful C–C and C–F covalent bonds, Perfluoro-polymers are resistant to many chemicals, which leads to these polymers being typical for applications that are submitted to hostile situations [30]. One of the drawbacks is their semi-crystalline nature and low solvent processability, which largely obstruct the development of the polymers in gas separation. In the mid-1980s, the gas permeation data were obtained through the introduction of many amorphous perfluoro-polymers with specific trade names, such as Teflon™ AF, Hyflon™ AD, and Cytop™. In Figure 2, the chemical structures of these perfluoro-polymers, which are commercially obtainable, are demonstrated. They are either cyclic homopolymers or copolymers of tetra-fluoro ethylene and perfluoro odioxole, where they are known for their elevated gas permeability because of the pre-existing micro channels [31]. The properties of these glassy perfluoro-polymers are shown in Table 3, listed from the most permeable Teflon™ AF2400 to the least permeable Cytop™.

### 2.3. Polymers of Intrinsic Microporosity (PIMs)

PIMs are a kind of glassy polymer with hard and twisted macromolecular backbone structures, which were originally reported by Budd and McKeown [38]. Unlike other porous organic polymers, PIMs are solution-treatable [38]. They are generated from the locations of twisting or spiro centers, where the poor molecular packing is induced by the restricted chain turnover of the component macromolecules, performing interconnected holes of smaller than 2 nm [39]. The substantial microporosity of this kind of polymer produces a less than 20% fractional free volume (FFV), resulting in elevated gas permeability [40]. The substantial microporosity of this kind of polymer produces a less than 20% fractional free volume (FFV), resulting in elevated gas permeability. These advantages swiftly stimulated concentrated research potential to assemble various PIMs to enhance the gas permeability and selectivity [38]. Table 4 represents the transport properties of the newly improved PIMs, involving spirobiindane (SBI)-based PIMs, Tröger’s base (TB)-based PIMs, polyimide (PI)-based PIMs, and some other differences. The crosslinking strategy to handle the fast physical aging is also summarized. 

The kinks of the PIMs polymer backbone were primarily recognized through inserting the SBI moiety with large pendant sets. SPI can be defined as a molecule with two indanes linked by a spiro carbon center. The SPI piece is generally polymerized with a halogen-including aromatic monomer, which results in the first PIMs used in gas separation membranes, such as PIM-1 and PIM-7 [41,42]. In PIMs, the gas permeation follows the solution-diffusion mechanism. The CO_2_/H_2_ selectivity is around 1-3, which is due to the size sieving trait that prefers the diffusion of H2. A spiro-fluorene (SBF) element could replace the SBI center, resulting in PIM-SBF offering low chain flexibility and an ultrahigh CO_2_ permeability of 13,900 Barrers [43]. Recently, big tetra methyl tetra hydro naphthalene (TMN) units were fused into the SBI units in PIM-1, leading to less conformational flexibility and resulting in higher CO_2_ permeability of 17,500 Barrers [44]. 

Unlike the SBI-based PIMs, Tröger-base (TB), a hard bicyclic unit, has been applied to PIMs. Diamino aromatic polymers with bicyclic rings, such as ethano-anthracene (EA) and triptycene (Trip), are integrated with or without SBI centers to compose an extremely twisted network. The TB group supplies extra Langmuir affinity toward CO_2_, which enhances CO_2_ selectivity [45]. The contortion sites can be within either the diamine [46] or the dianhydride [47]. Regardless of the utilization of different chemistry for PIM assembly, various enhanced PIMs have been notified by integrating bulky side groups including aromatic rings, such as the hexaphenylbenzene (HPB) unit that is used to reduce physical aging [48]. Carta et al. notified the CO_2_ permeability of 333,000 Barrers with a CO_2_/N_2_ selectivity of 14.9 through fusing bulky TMN and Trip into PIM [44]. Similar to other glassy polymers that have high free volume, PIMs suffer from physical aging, where the repose of the nonequilibrium series leads to the damage of permeability over time [49]. One intensively studied process is crosslinking to provide a more solid polymer network. For PIM-1, different crosslinking methods are used, e.g., thermal, UV, and chemical crosslinking, which are explained in detail in [50,51,52]. Driven by the high cost of the membrane, some polymeric materials have been blended with PIMs, such as carboxylate PIM-1 in Ultem and Matrimid as extremely permeable nanofillers, to mitigate the manufacturing cost as well as raise the gas selectivity [53,54]. Jue et al. found a defect-free HF asymmetric PIM-1 membrane created by phase inversion. A skin layer was provided, and CO_2_ permeance of 360 GPU and 27.7 CO_2_/N_2_ selectivity were gained [55].

**Table 4 membranes-12-00904-t004:** Transport properties of specific PIMs.

Strategy	Material	P (CO_2_)/atm	T/°C	CO_2_ Permeability/Barrer	(CO_2_/N_2_) Selectivity	(CO_2_/H_2_) Selectivity
SBI-based PIMs	PIM-1 alcohol-treated [41]	1	30	11,200	18.4	3.4
	PIM-7 [42]	0.2	30	1100	26	1.3
	PIM-SBF [43]	1	25	13,900	17.7	2.2
	PIM-TMN-SBI [44]	1	25	17,500	16.2	2.4
TB-based PIMs	PIM-EA-TB [56]	1	25	7140	13.6	0.92
	PIM-SBI-TB [42]	1	25	2900	12.5	1.3
	PIM-Trip-TB [57]	1	25	9709	15.9	1.2
PI-based PIMs	PIM-SBI-PI [47]	1	25	8210	18.7	3.1
	PIM-EA-PI [46]	1	25	7340	19.9	1.7
	6FDA-DAT1-OH [58]	2	35	47	25.9	0.37
PIMs wo SBI	PIM-TMN-Trip [44]	1	25	33,300	14.9	2
	PIM-HPB [48]	1	25	1800	20	7
Crosslinking	TOX-PIM-1 (thermal) [50]	4	22	5100	18.1	1.7
	PIM-1 (UV) [51]	4	22	6374	21.6	2.1
	PAH-PIM-1 (chemical) [52]	1	20	150	22.1	-
Blending	C-PIM-1/Matrimid [54]	3.5	35	2268	18.7	1.4
	PIM-1/Ultem [53]	3.5	35	3276	21.1	-
	PIM-1/POSS-PEG [59]	1	30	1309	31	-
	PIM-1/HCP [60]	2	25	19,086	11.6	-
Membrane	PIM-1 HF [55]	6.9	35	360 *	27.7	1

* GPU.

### 2.4. Thermally Rearranged (TR) Polymers

Thermally rearranged (TR) polymers are harder and planer macromolecules that can be composed by the thermal rearrangement of polyamide (PAs) or ortho-functionalized polyimides (PIs), which were first found by Park et al. [61]. Generally, the TR polymers are characterized by unprecedented polymer chain hardness and a tight bore size distribution, as a result of the microporous nature which occurs due to the high torsional energy fence versus turnover between the phenylene-heterocyclic circles. The main ancestors for the TR conversion are ortho-functionalized PIs or PAs, as shown in Figure 3 below. 

When the temperature rises above 350 °C, an intramolecular cyclization is started, and a hard polymer, such as polybenzoxazole (PBO), polybenzimidazole (PBI), or polypyrrolone (PPL), is composed if the ortho-functional set is amino or hydroxyl [62]. The TR polymers obtained from the polycondensation of hydroxy-diamine and diacid chloride are named TR-β, while the ones from the polycondensation of dianhydride and ortho-functional diamine are designated TR-α [62]. The transport properties of various selected TR polymers are shown in Table 5. 

### 2.5. Iptycene-Containing Polymers

The two molecules belonging to the iptycene family are triptycene (Trip) and pentiptycene (Pent), where the iptycene family is a group of three-dimensional molecules with arene identities integrated into the (2,2,2)bicyclooctatriene bridgehead method [77]. The iptycene membrane is unique due to the slits (blades) of the benzene creating an interior free volume, which can be compared to the kinetic diameters of the light gases in Table 1. The internal free volume, because of its shape and persisting nature, is not subjected to collapse, and low physical aging is obtained [78]. When the huge iptycenes are integrated into other polymer systems, the polymer chain packing is damaged and the overall free volume is increased [44]. Weidman et al. that proposed iptycene polymers can be classified into three categories: non-ladder, semi-ladder, and ladder, depending on the backbone architecture [79]. The categories are reviewed more systematically in Table 6.

## 3. Membrane Technology for CO_2_ Capture

To improve the CO_2_ capture system, several standards should be taken into consideration, such as high capture rate and low operating costs. Moreover, the flexibility of the membrane system plays the main role to choose the best configuration for the CO_2_ capture process. Presently, several procedures are utilized with different parameters to optimize CO_2_ capture technology. The membrane process requires a high separation of acid gases and impurities that are commonly part of the flue gas stream to avert harmful issues and extend its lifetime [88]. Figure 4 shows the membrane system used, with the acid gas separation process, which is integrated with a coal-fired power plant (CFPP).

The low volumetric fraction of CO_2_, contrasted with a high volume of the flue gas stream to be treated, is the essential challenge for the post-combustion capture process, which drives a low driving force of CO_2_ permeation. To overcome the low motive force in the membrane process fused into post-combustion CO_2_ capture technology, either a compressor before the membrane module or a vacuum pump in the permeate flow side, or both together, can be used [89]. As a result, the flue gas stream must be dried before entering the compression station to avoid troubles caused by water droplets.

In this paper, three different schemes with several parameters have been presented, either by using 1-single stage (with and without a vacuum pump) or 2-stages of the membrane to obtain both goals of 90% carbon capture rate and at least 95% purity of CO_2_ captured. High purity is required for the transportation purposes and for other goals such as enhance oil recovery (EOR) [90]. As can be noticed, the flue gases must be compressed before any membrane module to produce sufficient driving force for CO_2_ separation. As researched further, membrane surface area and energy consumption demonstrate important factors of any membrane module for CO_2_ capture. Designing the CO_2_ separation system utilizing membrane reveals the size of the membrane technology, such as the required surface area and the suitable configuration to achieve the CO_2_ purity goal (min. 95 mole %) with the lowest power consumption. In general, the efficiency of the process relies essentially on the compression ratio of the flue gas, membrane permeability, and membrane surface area. On the other hand, the purity of the CO_2_ stream depends on membrane selectivity, capture efficiency, and CO_2_ content in the combustion gases. This analysis concentrated on designing the optimal process with minimal energy and surface area needed to reach the proposed project targets.

In the research project (13/2020) a currently developed procedure was utilized to embed CA enzyme into polyacrylamide polymer (PSF 50 K) [91]. Generally, the lifetime of the membrane is 5 years, and after this period the membrane performance will decrease and the material must be replaced [8,92]. The permeability and selectivity for different gases are mentioned in Table 7. 

The objective of this article is to evaluate and compare the performance of a 1-single stage and 2-stages membrane unit fused into a conventional CFPP. The flue gas temperature and pressure treated in this study are 50 °C-1.015 bar, respectively (Table 7). Moreover, a high compressor pressure must be utilized in four stages (in this study we assumed it 70 bar) to compress and prepare CO_2_ captured for transportation goals with an inter-cooling process to reduce the high temperature generated from the high-pressure compressor [93] in order to accomplish all the demands for CO_2_ transport. 

**Table 7 membranes-12-00904-t007:** The main parameters of CFPP and membrane.

Parameters	Main Data
Fuel characteristics [94]	72.30% C, 4.11% H, 1.69% N, 7.45% O,
	0.56% S, 13.89% ash; Moisture: 8%;
	Lower heating value: 28,141 kJ/kg
CFPP parameter	
Steam temperature/pressure, [°C/bar]	585/290
Efficiency in high/medium/low-pressure steam turbine, [%]	84.9/91.6/87.8
Condensing pressure, [bar]	0.05
Heating water in the condenser, [°C]	9.5
Coal combustion efficiency in the steam generator, [%]	91
Steam flow rate, [t/h]	914.5
Net power plant efficiency, [%]	45.78
Flue gas Parameters	
Temperature/pressure [°C/bar]	50/1.013
Flue gas flow, [kmol/h]	40,320
Flue gas content, [mole%]	
CO_2_	13.12
N_2_	80.80
O_2_	6.03
SO_2_	0.04
Membrane parameters [95]	
Membrane material characteristics	Spiral wound in counter current
CO_2_ permeance, [GPU]	1000
N_2_ permeance, [GPU]	20
CO_2_/N_2_ selectivity	50
Compressor/vacuum pump efficiency, [%]	90
Variations of different membrane parameters simulated	
1st Compressor pressure, [bar]	1.5–10
Vacuum pump pressure (case B), [bar]	0.5–0.05
2nd Compressor pressure (case C), [bar]	2–10
1st membrane surface, [m^2^]	100,000–1,000,000
2nd membrane surface (case C), [m^2^]	5000–100,000

Three configurations for integrating membrane technology in CFPP, where the variations for these configurations are demonstrated in the membrane parameters section of the Table 7, are proposed as follows:A.1-single membrane with a compression station before the membrane inlet, see Figure 5;B.1-single membrane with a compression station before the membrane and a vacuum pump on the permeate flow side, see Figure 6;C.2-stages of a membrane with different compressors and vacuum pumps, check Figure 7.

In general, for a specific CO_2_ capture efficiency, the operation at a high-pressure difference across the membrane shows more power consumption required, a smaller membrane surface area, and higher CO_2_ purity until specific pressure. Furthermore, the performance at a lower pressure difference implies lower energy consumption, larger membrane surface area, and lower CO_2_ purity.

## 4. Decarbonized Coal-Based Super-Critical Power Plants, Main Design Characteristics, and Assessment Methodology

As a targeted industrial process to be decarbonized by membrane systems, a 330 MW net output coal power plant was investigated, using lignite coal as a fuel. The decarbonization yield is 90% the same as most of the CCS projects [93]. As can be distinguished from Figure 4, the flue gas out of the coal combustion process must be subjected to particulate matter (NO_x_ and SO_x_) removal before the CO_2_ capture process.

Table 7 presents the main technical assumptions of investigated CFPP and membrane.

All processes were modeled and simulated by using ChemCAD software. The paper analyzed different parameters of the compressor pressure, membrane surface area, and vacuum pump pressure (if used) in many cases to estimate the techno-economic influence of the analyzed CO_2_ capture solutions and achieve the membrane efficiency of 90% and CO_2_ purity of 95% with at least the minimum energy consumption required. However, power consumption in membrane technology is only the energy required for the main driving machines such as compressors and vacuum pumps. 

To evaluate the techno-economic influence of the analyzed CO_2_ capture solutions, the following indicators have been proposed [96]:

SPECCA—specific primary energy consumption for avoiding CO_2_ emissions;
(1)$$\mathrm{SPECCA}=\frac{3600\cdot \left(\frac{1}{\eta_{\mathrm{CCS}}}-\frac{1}{\eta_{\mathrm{base}}}\right)}{P_{\mathrm{base}}-P_{\mathrm{CCS}}},$$
where $\eta_{\mathrm{base}}$: the overall efficiency of energy solutions without CCS technology,

$\eta_{\mathrm{CCS}}$: the overall efficiency of energy solutions with CCS technology,

$P_{\mathrm{base}}$: the CO_2_ pollutant in kg/kWh produced by the CFPP without capture technology,

$P_{\mathrm{CCS}}$: the CO_2_ pollutant in kg/kWh produced by the CFPP with capture technology.

SEPCCAs,m—takes into account the energy penalty $\left(\eta_{\mathrm{base}}-\eta_{\mathrm{CCS}}\right)$ of the energy solution because of the extra heat consumption demanded in the chemical absorption process (${\mathrm{SPECCA}}_{s}$—Equation (2)), and the electricity consumption required by the capture process using membranes (${\mathrm{SPECCA}}_{m}$—Equation (3)): (2)$${\mathrm{SPECCA}}_{s}=\frac{3600\cdot \left(\eta_{\mathrm{base}}-\eta_{\mathrm{CCS}}\right)}{\eta_{\mathrm{base}}\cdot P_{\mathrm{base}}-\eta_{\mathrm{CCS}}\cdot P_{\mathrm{CCS}}},$$
(3)$${\mathrm{SPECCA}}_{m}=\frac{3600\cdot \left(W_{\mathrm{base},\mathrm{net}}-W_{\mathrm{CCS},\mathrm{net}}\right)}{W_{\mathrm{base},\mathrm{net}}\cdot P_{\mathrm{base}}-W_{\mathrm{CCS},\mathrm{net}}\cdot P_{\mathrm{CCS}}},\;$$
where $W_{\mathrm{base},\mathrm{net}}$: the net power generated for the energy solution without capture solution; $W_{\mathrm{CCS},\;\mathrm{net}}$: the net power generated for the energy solution with capture solution.

The levelized cost of electricity (LCOE) was calculated by Equation (4), considering the annualized CAPEX and OPEX costs, a specified CO_2_ recovery ($\varphi_{\mathrm{CO}2}),$ and the CO_2_ recovery flow (${\mathrm{CO}}_{2\mathrm{CCS}})$, where 6570 demonstrates 75% of annual capacity of hours:(4)$$\mathrm{LCOE}=\left(\frac{\mathrm{CAPEX}+\mathrm{OPEX}}{6570\cdot \varphi_{\mathrm{CO}2}\cdot {\mathrm{CO}}_{2\mathrm{CCS}}}\right)$$

Both recovery and avoided costs of CO_2_ indicators are calculated by taking the consideration LCOE, ${\mathrm{CO}}_{2\mathrm{CCS}}$, and specific CO_2_ emissions for energy solution with and without CCS solution:(5)$${\mathrm{CO}}_{2\mathrm{rc}}=\frac{{\mathrm{LCOE}}_{\mathrm{CCS}}-{\;\mathrm{LCOE}}_{\mathrm{base}}}{{\mathrm{CO}}_{2}\;\mathrm{recovery}}$$
(6)$${\mathrm{CO}}_{2\mathrm{ac}}=\frac{{\mathrm{LCOE}}_{\mathrm{CCS}}-{\;\mathrm{LCOE}}_{\mathrm{base}}}{{\mathrm{CO}}_{2\mathrm{base}}-{\mathrm{CO}}_{2\mathrm{CCS}}}$$

In order to set the economic indicators, several data on the unit costs of the components are presented in Table 8. The presented costs are calibrated to the year 2022 based on the index demonstrated on the Chemical Engineering online site [97].

In order to be able to determine whether an investment project, in this case, the CFPP with and without CO_2_ capture, is economically appropriate, an economic and financial analysis is required that considers all cash flows in and out of the established meter. The economic and financial indicators calculated in this analysis are as follows: 

Net present value (NPV) calculated with Equation (7):(7)$$\mathrm{NPV}=\sum_{i=1}^{n_{f}}\frac{{\mathrm{IN}}_{i}-C_{i}-A_{i}}{{\left(1+r\right)}^{i}}-\sum_{i=1}^{n_{r}}I_{i}\cdot {\left(1+r\right)}^{i},\;€$$
where ${\;\mathrm{IN}}_{i}$: the realized revenues for a year i (€/year); $C_{i}$: the operating and maintenance expenses for the year i, with taxes and duties but without depreciation (€/year); $A_{i}$: the annuity for the year i, if a loan was taken (€/year); $I_{i}$: the realized equity investment for the year i (€/year); r: the discount rate, which for the energy sector is 8%.

Internal rate of return (IRR) was determined utilizing Equation (8).
(8)$$\mathrm{NPV}=\sum_{i=1}^{n}\frac{{\mathrm{IN}}_{i}-C_{i}-I_{i}}{{\left(1+\mathrm{IRR}\right)}^{i}}=0,\;$$

IRR for an investment project is equal to the discount rate for which NPV is 0.

Discounted payback period (DPP) was determined by Equation (9).
(9)$$\mathrm{NPV}=\sum_{i=1}^{\mathrm{DPP}}\frac{{\mathrm{IN}}_{i}-C_{i}-I_{i}}{{\left(1+r\right)}^{i}},€$$

DPP is the period of time after which the initial investment is recovered.

The profitability index (IP) was calculated using Equation (10)
(10)$$\mathrm{IP}=\frac{\mathrm{NPV}+\mathrm{IA}}{\mathrm{IA}}$$
where IA: the discounted investment. An investment project is economically efficient if IP ≥ 1; for IP < 1 the project is economically inefficient. 

## 5. Results and Discussion 

In Case A (no vacuum pump used), various compressor pressures (1.5–10 bar) were used, while all other parameters were fixed. CO_2_ capture efficiency rose visibly with the increase of 1st compressor pressure (CP_1_), and the power consumption value rose as well. On the other hand, when CP_1_ was fixed and 1st membrane surface (SA_1_) differed from 100,000 to 1,000,000 m^2^, CO_2_ capture efficiency and power consumption increased significantly depending on the raising of CO_2_ captured. The 90% efficiency required for the process was obtained at 8.5 bar and 300,000 m^2^ SA_1_, while the energy consumption at this point was about 153 MW, and CO_2_ purity was 49%. However, the maximum CO_2_ purity achieved was 64% at 9.5 bar and 100,000 m^2^, which is quite low. The results show increasing the membrane surface area leads to a decrease in CO_2_ purity due to the other particles (e.g., N_2_) that pass through the membrane with CO_2_ molecules at a higher membrane surface. 

Case B demonstrates the utilization of vacuum pump (VP) pressures (0.05–0.5 bar) while the other parameters are constant. The results showed an obvious increase in CO_2_ capture efficiency when VP pressure decreased due to the high-pressure difference across the membrane unit. Thus, the power consumption value rose with the decrease of VP pressure. The 90% efficiency required for the process was obtained at 5.5 bar CP_1_, 200,000 m^2^ membrane SA_1_, and 0.15 bar VP pressure. At the same parameters, the CO_2_ purity was 68% and power consumption was about 145 MW. Moreover, the highest CO_2_ purity value was achieved at 2 bar CP_1_, 0.05 bar VP pressure, and 100,000 m^2^, which is 84%.

Since CO_2_ capture efficiency requires a large surface area and CO_2_ purity needs a low surface area to be high, 2-stages of membrane units have been recommended to manipulate and increase both the values of CO_2_ capture efficiency and CO_2_ purity (case C). 

In case C, different parameters of CP_1_ (2–10 bar) were used where all other parameters were constant. Consequently, the CO_2_ capture efficiency increased obviously with increasing CP_1_, and the power consumption rose as well. The 2nd compressor pressure (CP_2_) rose from 2 to 10 bar while fixing all other components, influenced and increased the 2nd membrane efficiency, and also impacted the CO_2_ purity. The moment when SA_1_ increased and all other parameters were constant, CO_2_ capture efficiency and power consumption rose excessively due to the CO_2_ captured rising. On the other hand, the leading factor that influenced the CO_2_ purity was the 2nd membrane surface (SA_2_), which increased from 5000 to 100,000 m^2^, where CO_2_ purity decreased constantly with the increase of the surface area. The efficiency and CO_2_ purity required for the process (90%, and 95%, respectively) were achieved at 8 bar CP_1_, 4 bar CP_2_, 600,000 m^2^ SA_1_, and 40,000 m^2^ of SA_2_. Moreover, the energy consumed in this case was around 189 MW, which is almost 57% of the total output of energy (330 MW). As shown in Figure 7, the flue gas exits from the second membrane has to be sent back to the mixer as a recirculated flue gas in order to increase CO_2_ capture efficiency.

The parameters (such as VP, SA_1_, and CP_1_) were selected and fixed for all the following figures only to illustrate the variations among the variants. 

In Figure 8, the influence of VP and CP_1_ on CO_2_ capture efficiency was analyzed. In the (no vacuum) line, the maximum capture efficiency value achieved was 87% at high CP_1_ (10 bar), while in 0.05 bar VP, the efficiency was much more than other efficiencies, reaching 99.9 % at 10 bar CP_1_ due to the high-pressure difference around the membrane. The impact of reducing VP on process efficiency was also clearly apparent. Moreover, it was also demonstrated that CO_2_ capture efficiency increases when CP_1_ increases successively because of the high CO_2_ content passed via the membrane unit.

Figure 9 presents how the CO_2_ purity differs based on the compressor and vacuum pump pressure. The necessity of utilizing a vacuum pump is clear to increase CO_2_ purity, where the highest amount achieved in no vacuum case is 62%. This value is poor compared with CO_2_ purity after using different vacuum pump pressures. On the other hand, increasing compressor pressure value leads to high CO_2_ purity till a specific value of the pressure where other molecules (such as N_2_) will pass through the membrane decreasing CO_2_ purity.

The impact of increasing vacuum pump pressure on CO_2_ capture efficiency at various CP_1_ is demonstrated below in Figure 10. The efficiency lines go down decreasingly at different CP_1_, as a result of the reduction of the pressure difference across the membrane. For example, CO_2_ capture efficiency at the point where CP_1_ and VP_1_ are 10, 0.5 bar, respectively, is higher than what is at 5.5 bar CP_1_ and VP_1_ of 0.05 bar due to the high pressure difference.

In Figure 11, the CO_2_ capture efficiency rises directly when the membrane surface increases in various CP_1_ values. At 10 bar CP_1_, it is evident that the efficiency over different membrane surface areas is almost steady at 100% due to the flow that is almost fully captured and being passed through the membrane. 

Figure 12 demonstrates the influence of the membrane surface at different compressor pressures. The power consumption increases continuously as the membrane surface increases at any CP_1_ (1.5–10 bar); it is also noticeable that the power consumption columns are taller when the CP_1_ rises due to the high gas flow that passed through the membrane unit. Typically, the power consumption required relies basically on the compressor and vacuum pump energy, and also on the power needed to compress the CO_2_ stream at 70 bar which increases constantly with the rise of the membrane surface.

In Figure 13, the impact of the SA_1_ with various CP_1_ on CO_2_ purity has been examined, where the purity reduced slightly at 1.5 bar because of the low compressor pressure value utilized, while at 5 bar the CO_2_ purity level was favorable in 100,000 m^2^ of SA_1_, almost 70% mole. As discussed before in the results section (case A), when the membrane surface increases the CO_2_ purity decreases constantly.

Figure 14 below shows the effect of CP_2_ on CO_2_ capture efficiency in different CP_1_. It is noticeable that CO_2_ capture efficiency increases crucially when the CP_1_ rises, reaching almost 100 % at 10 bar and CP_2_ at 4 bar. The figure also demonstrates that CP_2_ has a low impact on CO_2_ capture efficiency due to its location after the 1st membrane unit (see Figure 7). The explication of that tiny decrease in CO_2_ capture efficiency regarding CP_2_ is that the increase of CP_2_ leads to a rise in the 2nd membrane CO_2_ efficiency which drives a decrease in the recirculated flow to the mixer, finally reducing the flow rate entering the 1st membrane.

Figure 15 represents the influence of different SA_1_ on CO_2_ capture efficiency. The CO_2_ capture efficiency line goes up obviously with the increase of SA_1_ because of the high stream flow passed via the membrane at a higher membrane surface, obtaining almost 95% at 600,000 m^2^.

In Figure 16, the impact of CP_1_ on power consumption at different CP_2_ has been shown. The main factor that influences the total power consumption rate is CP_1_ because of the recirculated flue gas that fuses with the primary flue stream to generate a high flow rate that boosts the power needed to pressure the flow at the 1st compressor. Since the high CP_2_ pressures increase the 2nd membrane efficiency, the low recirculated flow decreases constantly, thus providing a lower flow rate entering the 1st compressor, which demonstrates why power consumption reduces at high CP_2_.

Figure 17 shows the effect of the 2nd membrane surface area on CO_2_ purity at different CP_2_. All CO_2_ purity lines go down permanently with the increase of membrane surface due to the transit of other components that pass through larger membrane surface (like N_2_) producing low CO_2_ purity. Furthermore, the CP_2_ affects CO_2_ purity, where higher CP_2_ leads to less CO_2_ purity. 

Figure 18 below represents the CO_2_ purity difference between the 1st and 2nd stages regarding various CP_2_. Firstly, it is distinguished that the CO_2_ purity of the 2nd membrane is much higher than that of the first because of the low surface area (20,000 m^2^) used in the 2nd membrane, which shows the significance of utilizing 2-stages of membrane unit. Typically, CO_2_ purity is reduced at high pressures. The influence of high CP_2_ on the 2nd membrane is more than in the first due to its location (see Figure 7). 

On the other hand, Figure 19 demonstrates the variation of CO_2_ purity of single-stage (with and without vacuum pump) and 2-stages membrane based on 1st compressor pressure. Low CO_2_ purity is remarkable in the 1-stage of the membrane (with and without a vacuum pump) due to the high membrane surface used (400,000 m^2^). Integrated 2-stages of membrane increase the CO_2_ purity with the rise of CP_1_ reaching almost 99% at 6 bar. It is observable that all the steam flow contents which passed through the 2nd membrane are CO_2_ molecules. However, the usage of the 2nd stage of the membrane is highly recommended to increase CO_2_ purity. 

Figure 20 exhibits how the CO_2_ capture efficiency of single-stage (with and without a vacuum pump) and the 2-stage membrane is affected by 1st compressor pressure. The CO_2_ capture efficiency line of single-stage with vacuum is remarkably higher than other lines, where the usage of a vacuum pump increases the pressure difference across the membrane unit, producing high CO_2_ capture efficiency.

Figure 21 demonstrates the impact of 1st compressor pressure on power plant efficiency at different 1-single stage membrane surfaces. As shown, high compressor pressure increases the energy consumption required for CCS, which drives to decrease the power plant efficiency. As described before, a high membrane surface such as 700,000 m^2^ increases CO_2_ capture efficiency, therefore, generating a significant demand for energy to compress the CO_2_ flow. Thus, the power plant efficiency decreases. 

Figure 22 shows a comparison of power plant efficiencies regarding 1-stage (no vacuum used), 1-stage (with vacuum), and 2-stages of the membrane. As mentioned above, high compressor pressure decreases the power plant efficiency either for the 1-single stage or 2-stages of the membrane. As shown, the power plant efficiency of 2-stages is higher than the efficiency of one stage due to the lower energy demanded for the 2-stage of membrane case. 

Figure 23 represents the difference in capital cost of various membrane surfaces of the 1-single stage based on different 1st compressor pressure. The capital cost lines of the surfaces increase and go up noticeably achieving 10,000 €/kWh at 10 bar of 700,000 m^2^. High surfaces raise the capital cost and also high compressor pressure influences the cost due to the high CO_2_ content passed through the membrane, which increases the power consumption required. 

The influence of SA_1_ on CO_2_ capture efficiency of one (with and without VP) and 2-stages of the membrane is shown in Figure 24. The case where the 1-single stage of the membrane with vacuum is used has the highest values of CO_2_ capture efficiency due to the high-pressure difference across the membrane unit. At 400,000 m^2^ of SA_1_, 1-stage (with vacuum) of membrane achieved almost 100% CO_2_ capture efficiency. While, in the same point, CO_2_ capture efficiency is less by around 6% and 13% for 1-stage (no vacuum) and 2-stages of the membrane, respectively. 

Figure 25 exhibits the variation of power plant efficiency of single-stage (with and without vacuum pump) and 2-stages of membrane regarding SA_1_. As noticed from the figure, at 200,000 m^2^, the power plant efficiency for 2-stages and no vacuum case of the membrane are almost the same, which are the highest, while utilization of vacuum pump in the 1-stage reduces the power plant efficiency by around 16% because of the low power consumption needed for capture CO_2_ comparing with 1-stage no vacuum case. By increasing SA_1_ the power plant efficiency lines of all stages go down constantly due to the increase of CO_2_ capture efficiency with SA_1_ increase, which leads to high demands of energy. 

Figure 26 represents the effect of SA_1_ on discounted payback period (DPP) of 1-stage (with and without vacuum pump) and 2-stages of the membrane. The SA_1_ of 1-stage (no vacuum) of the membrane has a low influence on DPP, which is the best case due to the low requirements of energy for CO_2_ capture compared with other cases. It is observable that increasing SA_1_ leads to an increase in DPP because of the high demand for power consumption. 

To evaluate the technical and economic assessment for the case A, the different parameters in Table 9 are chosen based on the optimum results of CO_2_ capture efficiency, CO_2_ purity, and power consumption (Figure 27). To abbreviate the surface area and compressor pressure parameters of case A, it was considered that case A_12_ represents 200,000 m^2^ of 1st membrane surface and 8 bar of 1st compressor pressure, and case A_13_ represents 200,000 m^2^ of 1st membrane surface and 10 bar of 1st compressor pressure. For 400,000 m^2^ SA_1_ and 6 bar of CP_1_, the abbreviation is A_21_ and along with others. 

Table 10 demonstrates the evaluation and economical estimation with the analyzed solutions considering the indicators presented above of single-stage membrane technology. By fusing the single-stage of the membrane without vacuum pump utilization, the net power plant efficiency decreases by (31–50%) based on the membrane surface and compressor pressure values utilized. Generally, increasing the membrane surface or compressor station unit drives an increase in the power plant efficiency loss. As demonstrated in the paper, the electrical consumption required for a membrane system increases vastly with the increase of CP_1_ in addition to membrane surface impact. The LCOE increases with the increase of membrane surface from 200,000 to 500,000 m^2^ of (44–56%) at compressor pressure of 8 bar. The CO_2_ avoided cost is higher on the 200,000 m^2^ membrane surface than on other surfaces at the same compressor pressure due to the low power required to capture CO_2_. 

By presuming the CO_2_ tax is 82 €/ton and retail electricity cost is 160 €/MWh and considering that all carbon certificates are sold, the assumptions for all cases are in the Tables below. The economic evaluation of the 1-single stage of the membrane (no vacuum used) was analyzed in Table 11. As summarized, by increasing the membrane surface from 200,000 to 500,000 m^2^ at the same compressor pressure, the net present value increases at 400,000 m^2^ by around 25%, then reduces by 1% at 500,000 m^2^ membrane surface due to the increase of energy consumption needed for the CO_2_ capture process.

To evaluate the technical and economic assessment for the case B, the different parameters in Table 12 are chosen based on the optimum results of CO_2_ capture efficiency, CO_2_ purity, and power consumption.

Table 13 summarizes the assessment with the examined solutions based on the indicators shown above at a 1-stage of membrane technology with vacuum pump usage. In all analyses, the vacuum pump pressure was assumed to be 0.25 bar. After the integration, the net power plant efficiency decreased by 42% at 200,000 m^2^ and 8 bar, then reduced constantly, reaching mitigation of 59% of the main net power plant efficiency at 500,000 m^2^. As explained in the paper, the high pressure difference across the membrane module guides an increase in the power plant efficiency loss. The table shows that LCOE increases by 23% with the increase of membrane surface from 200,000 to 400,000 m^2^ and by 16% with the increase from 400,000 to 500,000 m^2^ due to the high demands of energy CO_2_ capture. In terms of CO_2_ avoided cost, for the 200,000 m^2^ membrane surface the cost is less than 400,000 and 500,000 m^2^ by around 33% and 49%, respectively, at compressor pressure 8 bar because of the low energy required for the CO_2_ capture process.

Table 14 demonstrates the economic evaluation for different parameters which were examined. The net present value in the economic assessment decreases by 24% when the membrane surface increases from 200,000 to 400,000 m^2^, and also decreases if the surface differs from 400,000 to 500,000 m^2^ by around 21% at 8 bar CP_1_. On the other hand, the DDP scale increases with the membrane surface increase reaching 15.5 years at 500,000 m^2^ due to the high power consumption that increases with the rise of SA_1_.

To estimate the technical and economic assessment for the case C, the different parameters in Table 15 are chosen based on the optimum results of CO_2_ capture efficiency, CO_2_ purity, and power consumption.

Table 16 shows the technical estimation of 2-stages of membrane integrated into CFPP. As demonstrated, increasing SA_1_ drives the decrease of the net power plant efficiency by 14% when the surface is increased from 200,000 to 600,000 m^2^ at 6 bar of CP_1_. Integrating 2-stages of the membrane increases the LCOE by about 50% at 8 bar compressor pressure and 400,000 m^2^ membrane surface. By considering the impact of CP_1_ on CO_2_ avoided cost, the values of the cost decrease with CP_1_ increasing (6–10 bar) by around 66% at 200,000 m^2^ of SA_1_. 

In terms of economic assessment of 2-stages of the membrane, Table 17 summarizes it, where the net present value decreases with the increase of SA_1_ from 200,000 to 600,000 m^2^ by 59%. It is noticeable that increasing SA_1_ impacts directly on the DPP, where DPP decreases from almost 15 to 11 years with the rise of SA_1_ 200,000–600,000 m^2^. 

The sensitivity assessment for 1-stage, no vacuum, SA_1_ of 500,000 m^2^, CP_1_ 6 bar is given (as an example). For all cases, the variation is approximately the same. The influence of the CO_2_ tax and the CO_2_ capture efficiency on the levelized cost of electricity is remarkable. Taking into account that CO_2_ avoided cost is over 100 €/t for almost all cases, we calculated a possible 140 €/t of CO_2_ tax which allows us to increase the profitability index from 1.93 to 2.44.

Different economic parameters of 1- and 2-stages of the membrane are compared in Table 18 considering the optimal CO_2_ capture efficiency and purity. According to the table, increasing the membrane stages leads to an increase in CO_2_ purity, but at the same time increases the power consumption and investment cost of the project, which is also due to the usage of more equipment that requires energy to function. 

In order to provide a clear vision regarding modeling using the CHEMCAD process with membrane, a comparison between the current and other papers studied from technical and economical points of view has been presented in Table 19 below. 

The current paper focused on using membrane technology integrated into CFPP in the coming years (at least 5 years); in that time, regarding the improvements of pieces of equipment and performance, we believe the efficiency of equipment (such as compressors and vacuum pumps) could be enhanced. However, the difference between using a compressor or pump efficiency of 90% instead of 85% leads to 6% decrease in the energy consumption at the CO_2_ capture efficiency of 90%.

Study_1_ represents a paper studied by Xuezhong He and May-Britt Hägg [8], while study_2_ demonstrates a research article authored by Xuezhong He; Jon Arvid Lie; Edel Sheridan; and May-Britt Häg [99], study_3_ exhibits an article studied by Van der Sluus; Hendriks; and K. Blok [100], and study_4_ shows a paper studied by Merkel; Lin H; Wei X; and Baker [101]. The current study shows an increase regarding the economic side compared with study_1_ due to the elevated flue rate, which is almost 76% higher, which also explains why the power consumption of the present paper is larger as well. On the other hand, the capture efficiency and purity of CO_2_ is more than the others because of the high CO_2_ permeance and the 1st compressor pressure used. 

## 6. Conclusions

The aim of this paper was to assess and compare 1-single stage (with and without usage of vacuum pump) and 2-stages of membrane performance, from a technical and economical point of view, integrated into a 330 MW coal-based super-critical power plant with different configurations and parameters to achieve 90% CO_2_ capture efficiency and CO_2_ purity of at least 95%.

The application of a 1-single stage of the membrane with and without a vacuum pump station is remarkably undesirable due to neither the low CO_2_ capture efficiency and CO_2_ purity when no vacuum is used nor the poor CO_2_ purity when a vacuum station is harnessed. The required CO_2_ capture efficiency (90%) is obtained at 8.5 bar CP_1_ and 300,000 m^2^ of SA_1_ when no vacuum is utilized. At this point, CO_2_ purity attained was 50%, which is extremely low. The moment a vacuum pump is utilized, CO_2_ capture efficiency and CO_2_ purity can be improved. A 90% CO_2_ capture efficiency can be obtained at 4.5 bar CP_1_ and 200,000 m^2^ of SA_1_. At the same point, the power consumption is 10% less than no vacuum station used at 90% capture efficiency. The fundamental obstacle that has been examined is the low value of CO_2_ purity at any parameters utilized (84% mole max); therefore, a 2nd stage of the membrane with lower surface areas has been suggested to enhance the CO_2_ purity. The results show that CO_2_ purity has increased by almost 16% reaching 97% at 6 bar CP_1_, 4 bar CP_2_, SA_1_ of 600,000 m^2^, and SA_2_ of 40,000 m^2^. On the other hand, CO_2_ capture efficiency at this point is around 93%, which is the required level. However, high CP_1_ is a considerable factor in boosting the carbon capture rate. By increasing the CP_1_ from 4 to 6 bar, CO_2_ capture efficiency rises by around 51%, and this value has been obtained at 600,000 m^2^ of SA_1_. The SA_1_ impacts the CO_2_ capture rate as well, for example increasing the SA_1_ from 200,000 m^2^ to 600,000 m^2^ drives a rise in CO_2_ capture efficiency by approximately 31%. At the suitable parameters to achieve 93% and 97% of CO_2_ capture efficiency and CO_2_ purity, respectively, the power consumption required for the process is around 57% of the total plant energy (330 MW). 

One of the main influences on the economic section is SA_1_, where increasing the surface drives an increase in CO_2_ captured constantly which means higher electrical energy required. Moreover, CP_1_ has a senior role in affecting the economic side, where high CP_1_ leads to an increase in investment cost. The LCOE is highly influenced by increasing CP_1_, where raising CP_1_ 2–6 bar leads to an increase of 42% at the 2-stages of the membrane and 600,000 m^2^ SA_1_. However, at the optimum point where 93% and 97% of CO_2_ capture efficiency and CO_2_ purity, respectively, has achieved, the economic assessments are credible from DPP, CO_2_ avoided, and LCOE points of view. In applicable future growth methods, the increase of CO_2_ permeability is a significant factor to consider.

## Figures and Tables

**Figure 1 membranes-12-00904-f001:**
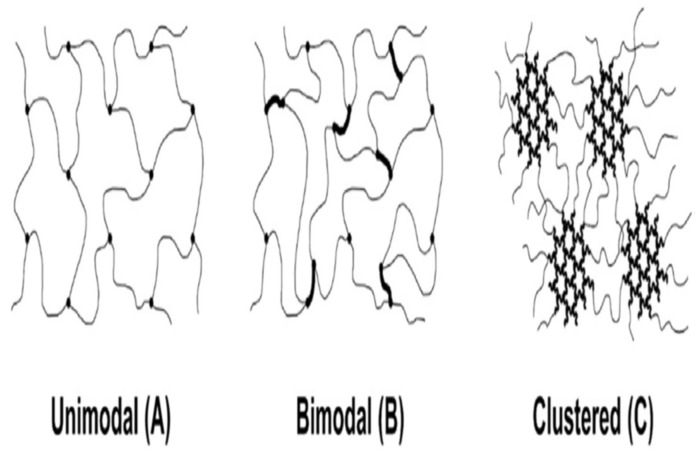
Schematics of unimodal, bimodal, and clustered crosslinked PEO networks. Adapted from [27].

**Figure 2 membranes-12-00904-f002:**
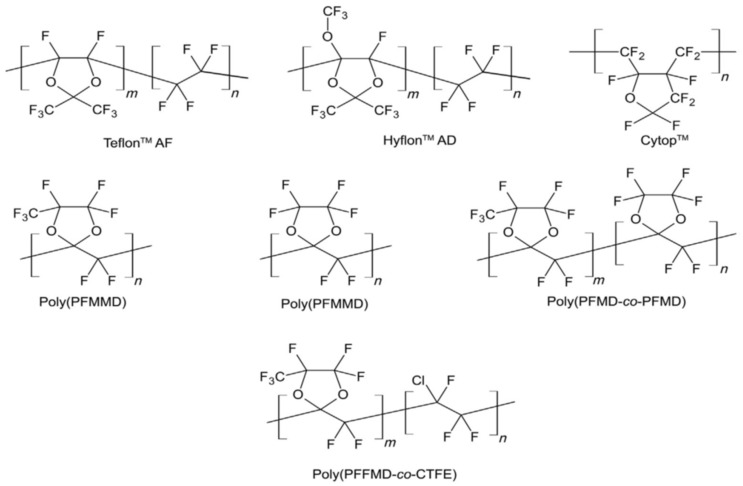
Perfluoro-polymers structures. Adapted from [11].

**Figure 3 membranes-12-00904-f003:**
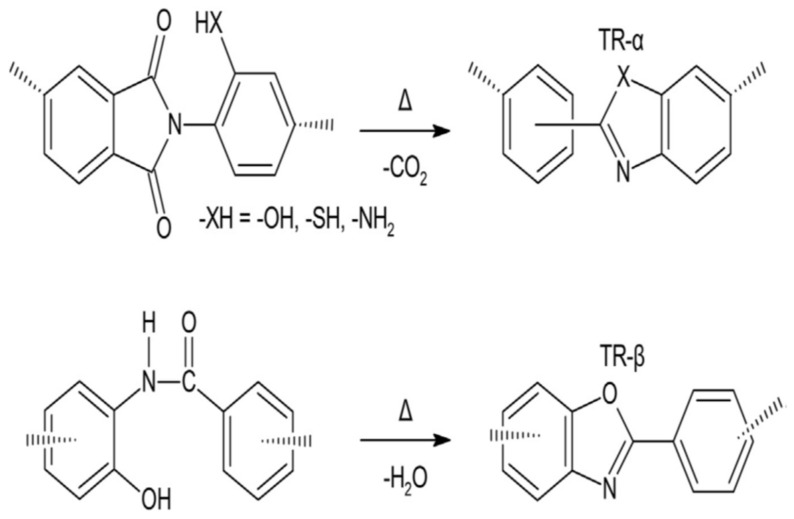
Thermal rearrangement of (1) TR-α, where the ancestor is an ortho-functional polyimide (PI), and (2) TR-β, where the ancestor is an ortho-functional polyamide (PA) [62].

**Figure 4 membranes-12-00904-f004:**
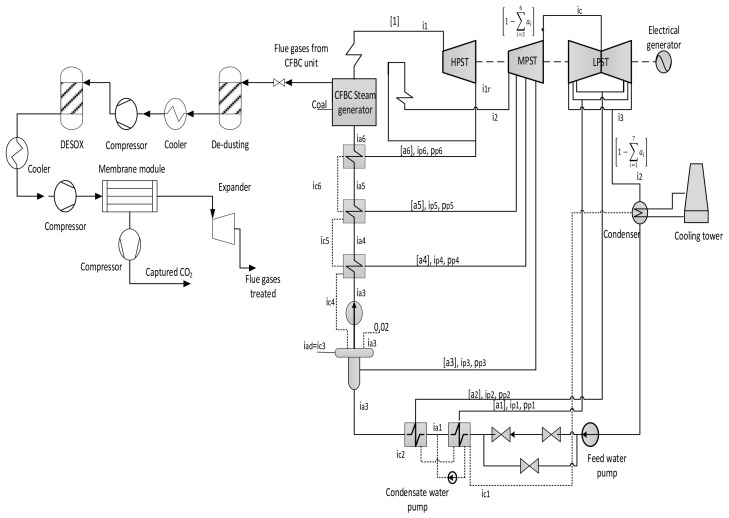
Principle diagram of membrane integration in CFPP.

**Figure 5 membranes-12-00904-f005:**
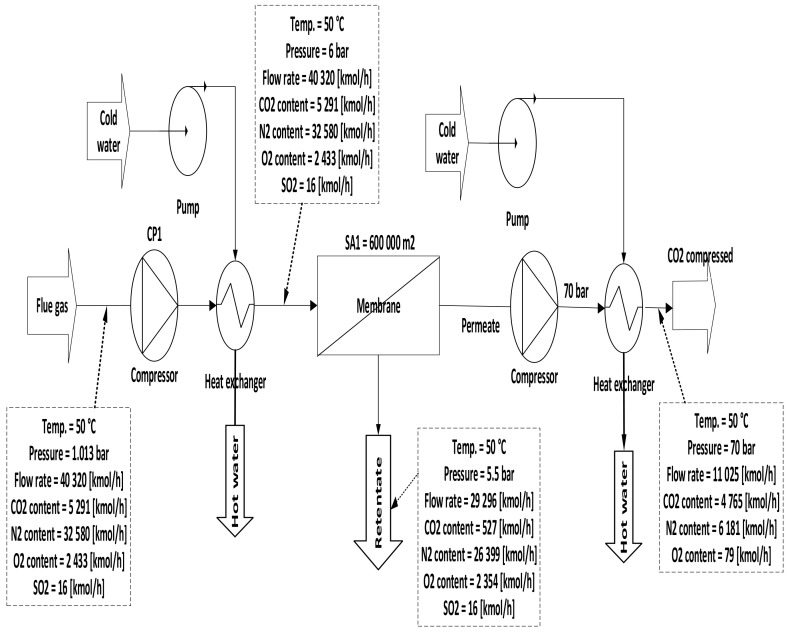
Single-stage scheme of a membrane with only a compression station.

**Figure 6 membranes-12-00904-f006:**
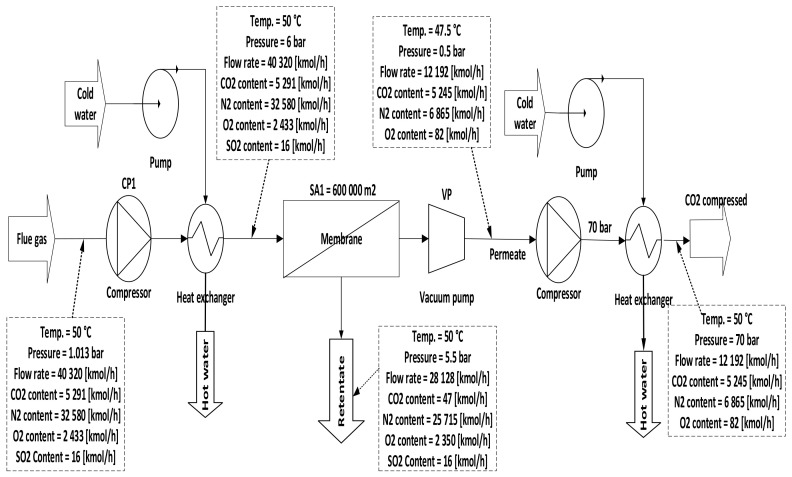
Single-stage scheme of a membrane with a compressor station and a vacuum pump.

**Figure 7 membranes-12-00904-f007:**
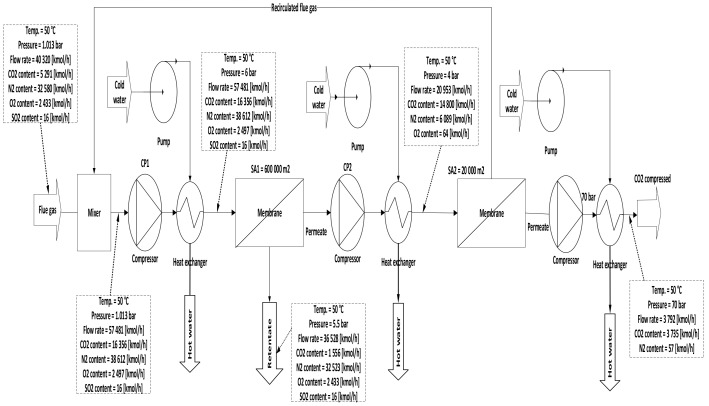
Two stages of a membrane with different compression stations.

**Figure 8 membranes-12-00904-f008:**
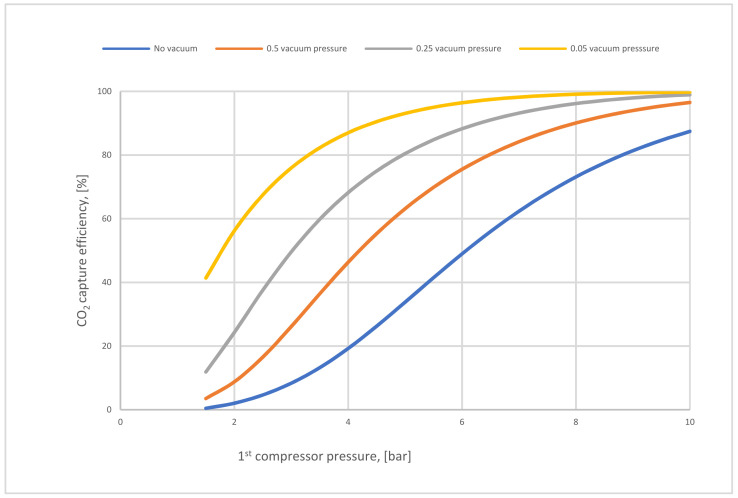
CO_2_ capture efficiency regarding different 1st compressor and vacuum pump pressure values for 1-stage (case A, B) at 200,000 m^2^ of 1st membrane surface.

**Figure 9 membranes-12-00904-f009:**
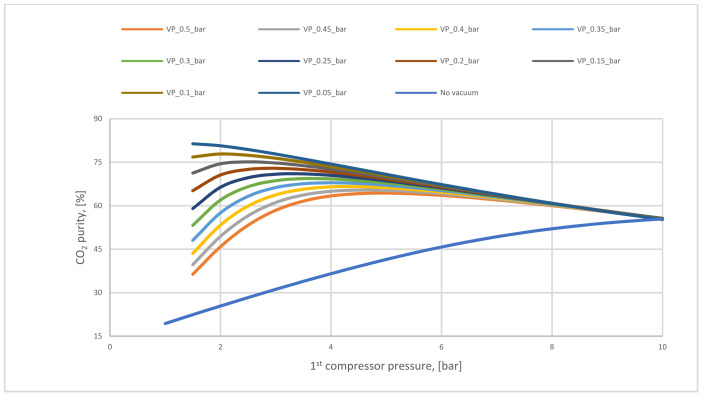
CO_2_ purity variation regarding vacuum pump for 1-stage (case A, B) at different 1st compressor pressures and 200,000 m^2^ of 1st membrane surface.

**Figure 10 membranes-12-00904-f010:**
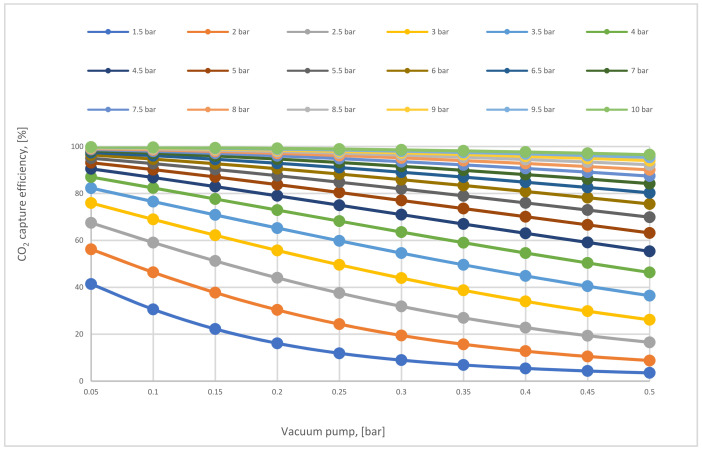
The vacuum pressure difference influence on CO_2_ capture efficiency for 1-stage (case A, B) at 200,000 m^2^ of 1st membrane surface.

**Figure 11 membranes-12-00904-f011:**
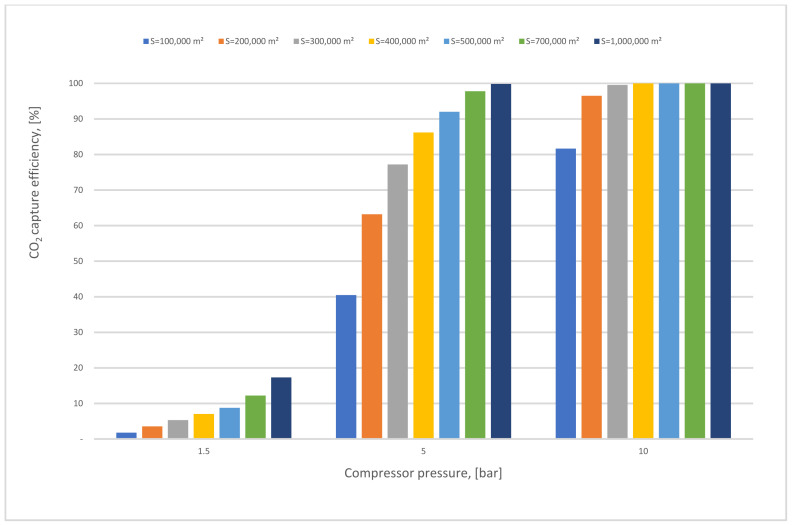
First membrane surface and 1st compressor pressure influence on CO_2_ capture efficiency for 1-stage (case A, B) at 0.5 of vacuum pump pressure.

**Figure 12 membranes-12-00904-f012:**
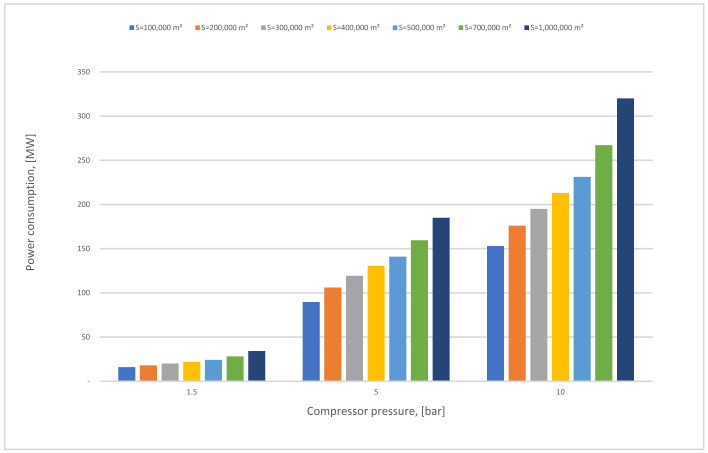
First membrane surface and 1st compressor pressure influence on power consumption for 1-stage (case A, B) and 0.5 bar of vacuum pump pressure.

**Figure 13 membranes-12-00904-f013:**
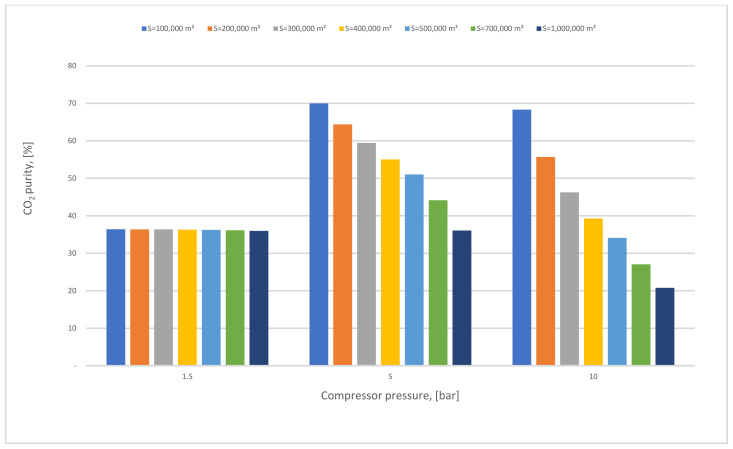
The effect of 1st membrane surface with different 1st compressor pressure on CO_2_ purity for 1-stage (case A, B) and 0.5 bar of vacuum pump pressure.

**Figure 14 membranes-12-00904-f014:**
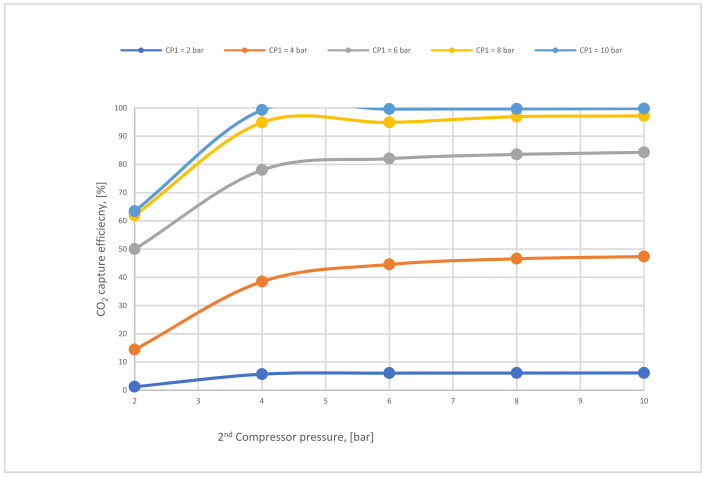
The impact of 2nd compressor pressure on CO_2_ capture efficiency for 2-stages at 600,000 and 40,000 m^2^ (1st membrane surface and 2nd membrane surface, respectively).

**Figure 15 membranes-12-00904-f015:**
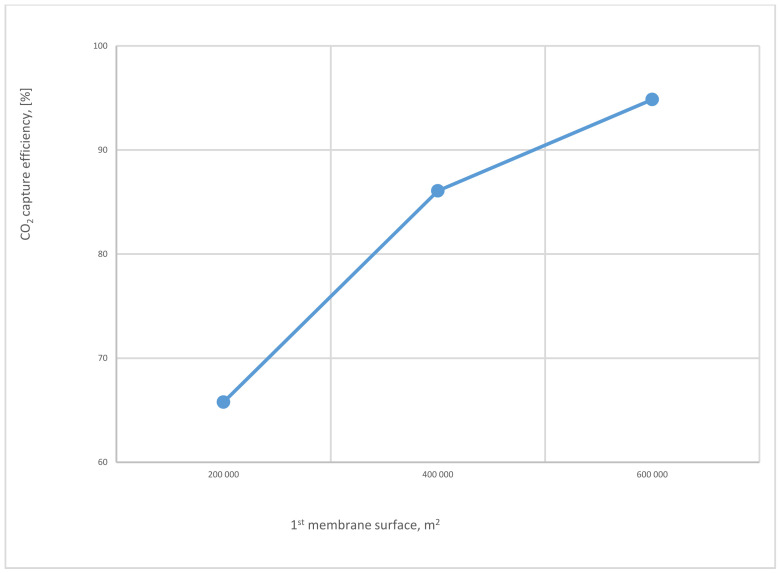
The 1st membrane surface effect on CO_2_ capture efficiency for 2-stages of the membrane at 8 bar of 1st compressor pressure, 4 bar of 2nd compressor pressure, and 40,000 m^2^ of 2nd membrane surface.

**Figure 16 membranes-12-00904-f016:**
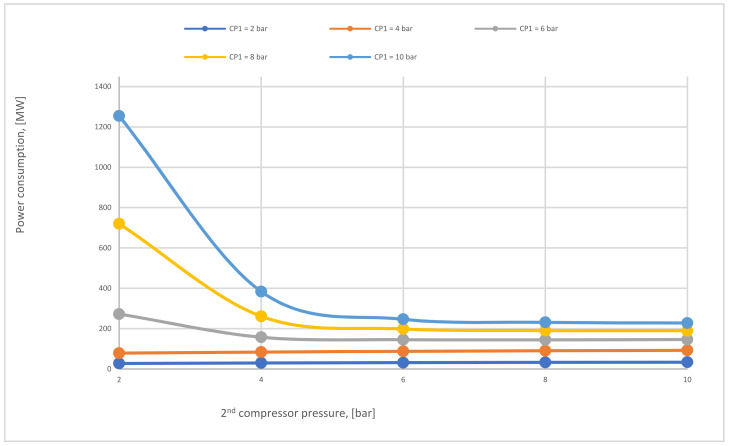
The total power consumption required for 2-stages of the membrane (case C) regarding different 1st and 2nd compressor pressures at SA_1_ of 600,000 m^2^ and SA_2_ of 20,000 m^2^.

**Figure 17 membranes-12-00904-f017:**
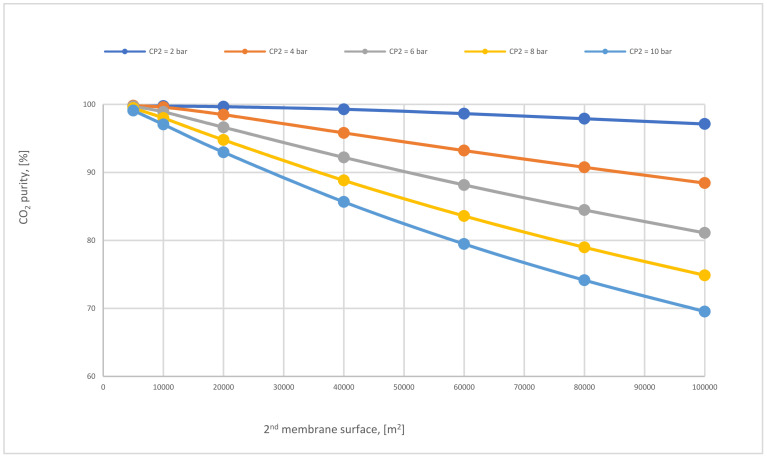
CO_2_ purity at different 2nd compressor pressures for 2-stages of the membrane regarding 2nd membrane surface at SA_1_ of 600,000 m^2^ and CP_1_ of 6 bar.

**Figure 18 membranes-12-00904-f018:**
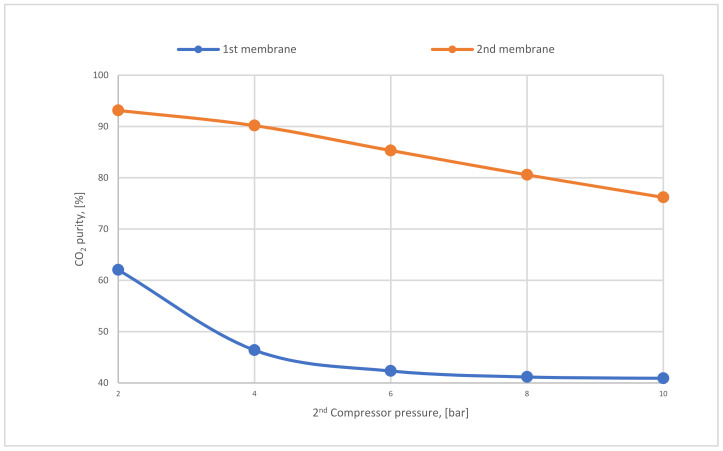
The difference in CO_2_ purity at CP_1_ of 4 bar, SA_1_ of 600,000 m^2^, and SA_2_ of 20,000 m^2^ regarding various 2nd compressor pressures.

**Figure 19 membranes-12-00904-f019:**
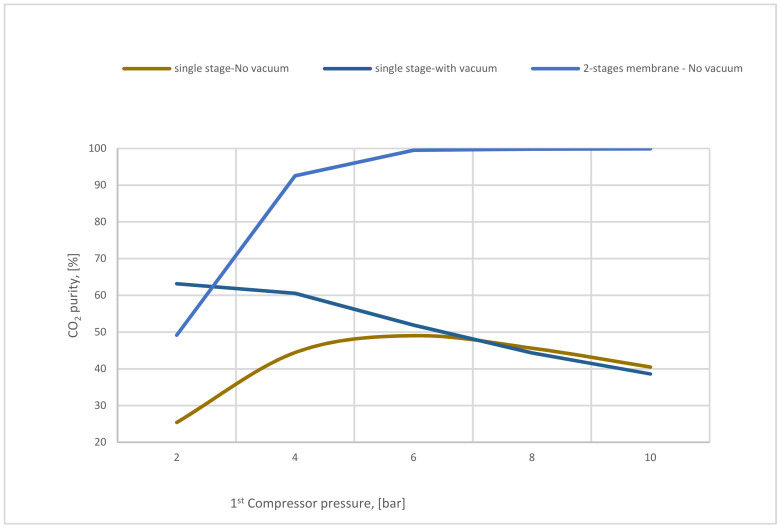
The variation of CO_2_ purity at 400,000 m^2^ of SA_1_ and 5000 m^2^ of SA_2_ based on 1st compressor pressure.

**Figure 20 membranes-12-00904-f020:**
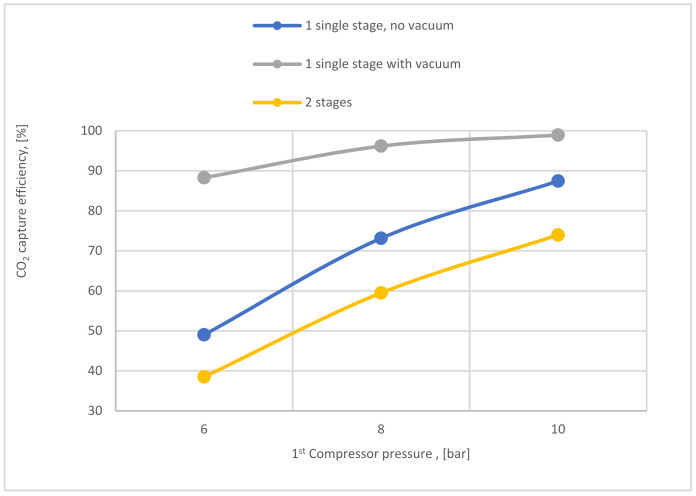
CO_2_ capture efficiency variation at 200,000 m^2^ of SA_1_, 0.25 bar of VP pressure, CP_2_ of 4 bar, and SA_2_ of 20,000 m^2^ based on different 1st compressor pressure.

**Figure 21 membranes-12-00904-f021:**
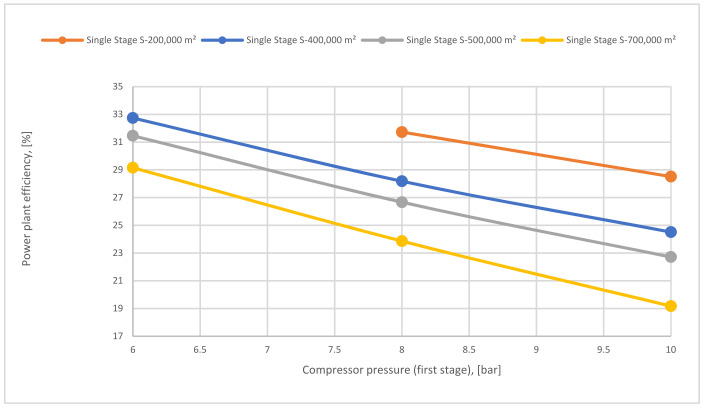
Power plant efficiency depending on 1st compressor pressure and different 1st membrane surfaces.

**Figure 22 membranes-12-00904-f022:**
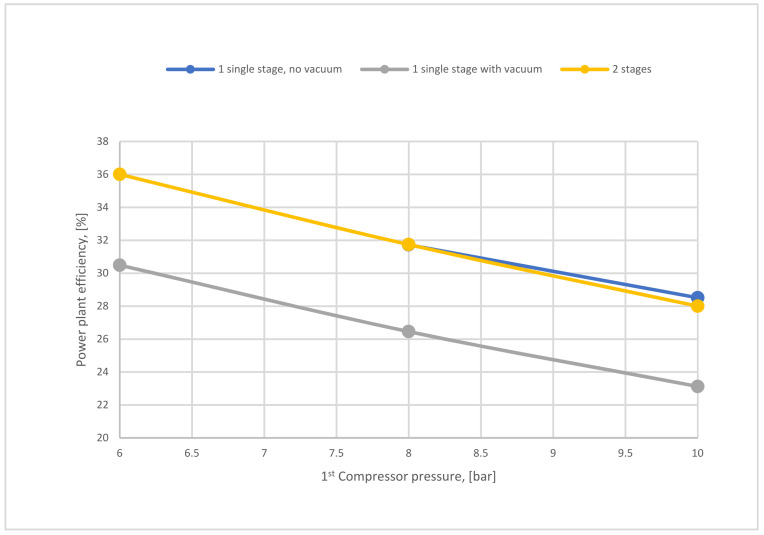
Power plant efficiency variation of 1-single stage (with and without VP) depending on 1st compressor pressure at 200,000 m^2^ SA_1_, 40,000 m^2^ SA_2_, and 4 bar of CP_2_.

**Figure 23 membranes-12-00904-f023:**
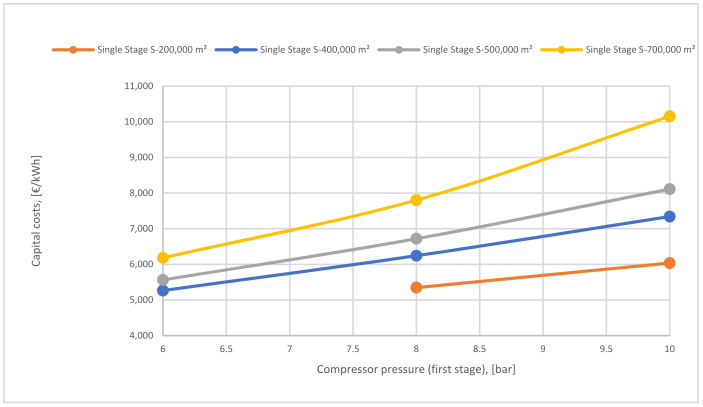
1st compressor pressure effect on the capital cost at different surfaces area.

**Figure 24 membranes-12-00904-f024:**
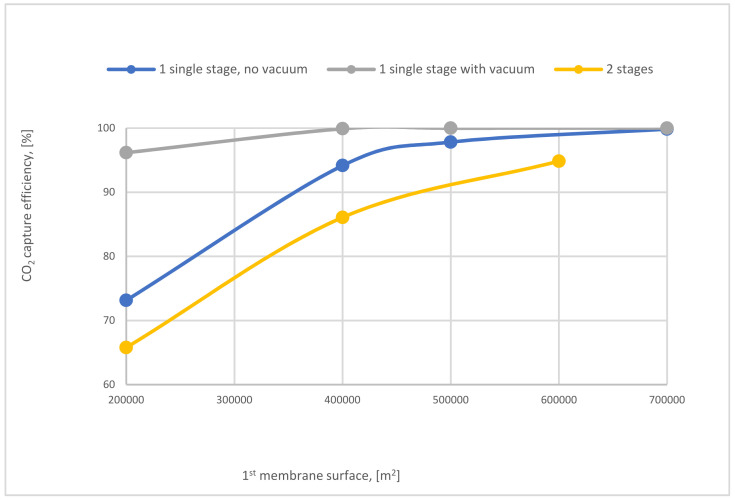
The impact of 1st membrane surface on CO_2_ capture efficiency of 1 and 2-stages of membrane at 8 bar CP_1_, 4 bar CP_2_, and 40,000 m^2^ of SA_2_.

**Figure 25 membranes-12-00904-f025:**
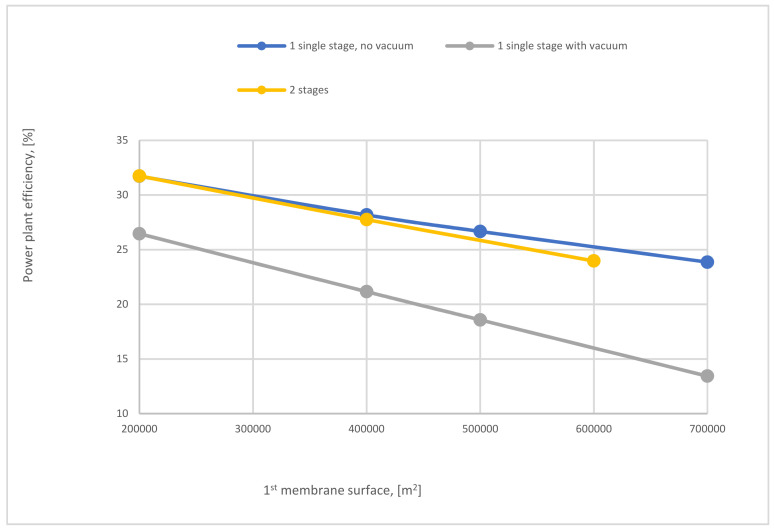
The impact of 1st membrane surface on power plant efficiency of 1 and 2-stages of membrane at 8 bar CP_1_, 4 bar CP_2_, and 40,000 m^2^ of SA_2_.

**Figure 26 membranes-12-00904-f026:**
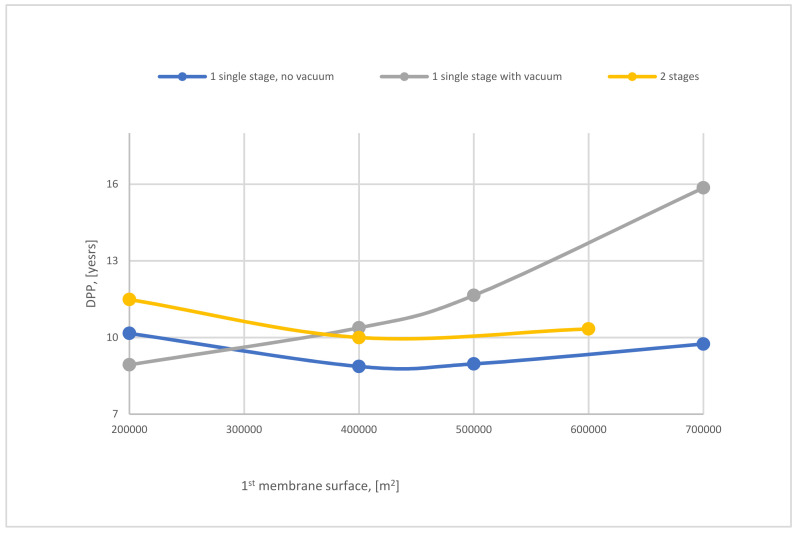
The variation of DPP of 1-stage (with and without VP) and 2-stage of the membrane at 8 bar CP_1_, 4 bar CP_2_, and 40,000 m^2^ of SA_2_.

**Figure 27 membranes-12-00904-f027:**
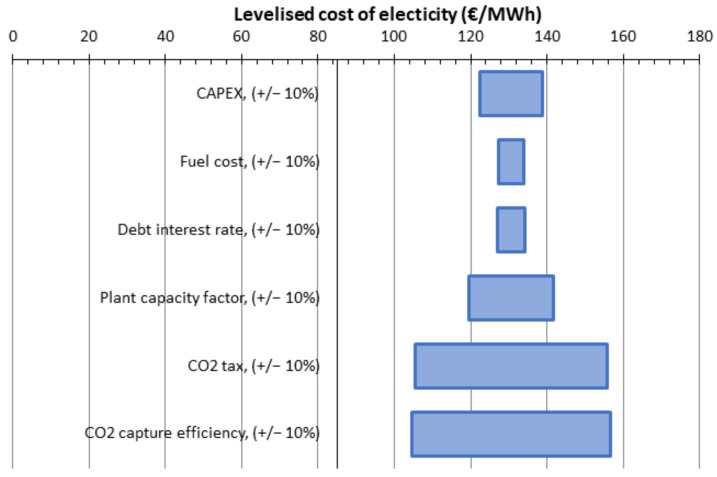
LCOE variation according to different parameters.

**Table 1 membranes-12-00904-t001:** Physical properties of different gases [14].

Gas	Kinetic Diameter/nm	Critical Temperature/°C
CO_2_	0.330	304.1
N_2_	0.364	126.2
H_2_	0.289	33.2
CH_4_	0.380	190.6

**Table 2 membranes-12-00904-t002:** Transport properties of specific PEO polymers.

Strategy	Material	T/°C	CO_2_ Permeability/Barrer	(CO_2_/N_2_) Selectivity	(CO_2_/H_2_) Selectivity
Copolymer	PEO-b-PA6 [14]	35	120	51.4	9.8
	PEO-b-PBT [20]	30	150	51.5	10.3
	PEO-ran-PPO T6T6T [21]	35	470	43	10
	Pent-PI-PEO2000 [22]	35	39	36	4.1
	PEO-b-PBT on PDMS [23]	30	1815 *	50	-
	PEO-b-PS [24]	70	20,400 *	27.7	-
Blending	PEO-PBT/PEG200 [20]	30	208	48.7	11.6
	PEO-PBT/PEG-BE [20]	30	400	50.1	11.8
	PEO-PBT/PEG-DBE [20]	30	750	40	12.4
	PEO-PPO-T6T6T/PDMS-PEG [25]	45	896	36	10.6
Crosslinking	PEO-526/dopamine/PEGDME [26]	50	200	30	6
	PEO-amine/PEO-epoxy [27]	35	376	53	10

* GPU.

**Table 3 membranes-12-00904-t003:** Transport properties of specific perfluoro-polymers.

Strategy	Material	P (CO_2_)/atm	T/°C	CO_2_ Permeability/Barrer	(CO_2_/N_2_) Selectivity	(CO_2_/H_2_) Selectivity
Commercial	Teflon™ AF2400 [32]	1	35	2200	4.6	0.96
	Teflon™ AF1600 [33]	1	35	520	4.7	1.06
	Hyflon™ AD80 [34]	3	35	473	6.1	1.19
	Hyflon™ AD60 [34]	3	35	124	7.3	1.63
	Cytop™ [34]	1	35	35	7	1.69
	Teflon™ AF2400 [35]	4.4	22	13,000 *	4.8	0.81
	Hyflon™ AD60 [35]	4.4	22	1330 *	7.3	1.28
Homo-polymer	Poly(PFMD) [36]	4.4	35	5.9	8.3	8.47
	Poly(PFMMD) [36]	4.4	35	58	7.5	4.1
Copolymer	Poly(PFMMD-co-PFMD) [37]	4.4	22	403 *	9.1	2.9
	Poly(PFMMD-co-CTFE) [37]	4.4	22	44 *	9	5.7

* GPU.

**Table 5 membranes-12-00904-t005:** The transport properties of various selected TR polymers.

Strategy	Material	P (CO_2_)/atm	T/°C	CO_2_ Permeability/Barrer	(CO_2_/N_2_) Selectivity	(CO_2_/H_2_) Selectivity
TR-α-PBO	6FDA + bisAPAF [61]	10	35	4 045	25.9	1.4
	6FDA + bisAPAF [63]	1	25	4 201	14.8	1
	6FDA + bisAPAF + ADHAB [64]	3	30	151	21	-
	6FDA + HAB [65]	1	35	2.9	29	0
TR-α-PBI	6FDA + DAB [66]	1	25	1 624	26.2	0.91
TR-β-PBO	BPDC + bisAPAF [67]	10	35	532	17.6	1
Crosslinking	6FDA + bisAPAF + DABA/diol [68]	1	25	746	25.2	1.2
	6FDA + bisAPAF + DABA [69]	1	25	491	24.3	1
Copolymer	6FDA + bisAPAF + DAM [70]	0	35	137	21.6	0.78
	6FDA + HAB + 4MPD [71]	10	35	226	10.5	-
TR-labile PI	6FDA + DABA + βCD [72]	10	35	2 707	15.3	0.34
	6FDA + durene + DABA + γCD [73]	2	35	1 024	18.2	0.24
TR w/SBI	TR-PIM-1 [74]	1	35	675	23	1.6
	TR-PIM-2 [74]	1	35	263	24	1
TR HF	6FDA + bisAPAF [75]	1	25	2 326 *	20	1.2
	6FDA + bisAPAF [76]	1	25	2 500 *	16	1.2

* GPU.

**Table 6 membranes-12-00904-t006:** Transport properties of specific iptycene-containing polymers.

Strategy	Material	P (CO_2_)/atm	T/°C	CO_2_ Permeability/Barrer	(CO_2_/N_2_) Selectivity	(CO_2_/H_2_) Selectivity
Non-ladder	6FDA + DATRI [80]	1	35	189	23.3	0.73
	6FDA + DAT2 [49]	2	35	210	23.3	0.74
	6FDA + PPDA(CF3) [81]	1	35	132	19	0.7
	TPDAn + 6FAP [82]	1	35	4.7	25	0
	TPHA-TC [83]	11	35	270	-	-
Semi-ladder	KAUST-PI-1 [84]	2	35	2 389	22.3	0.6
	KAUST-PI-2 [84]	2	35	2 071	21.1	0.87
	6FDA + PAF [85]	2	35	6.8	-	-
	TPDA + APAF [85]	2	35	46	-	-
	PBIBI + PPD [86]	1	25	137.2	27.8	-
Ladder	PIM-Trip-TB [87]	1	25	9 709	15.4	1.2
	PIM-Btrip-TB [87]	1	25	13 200	14.2	1.3

**Table 8 membranes-12-00904-t008:** Main assumptions concerning the economic indicators.

Indicators	Main Data
Availability factor, %	85
Electricity price, €/MWh	160
CO_2_ tax, €/t [98]	82
Annual hours, h/year	7446 (85/100·8760)
Membrane capture unit	
Membrane-specific cost, €/m^2^	50
Membrane lifetime, years	5
Flue gas, and inter-stage compressor, €/kW	850
Vacuum pump, €/kW	1300
CO_2_ pump, €/kW	1350
CO_2_ compressor, €/kW	1800
Membrane replacement cost	20% of the membrane cost
Labor cost, €/h	15
CO_2_ stream compression	
CO_2_ compressor, M€	11.7
Compressor inter-stage coolers and separators, M€	0.87

**Table 9 membranes-12-00904-t009:** The specific parameters were chosen for the technical and economical assessment of a single-stage of membrane technology (no vacuum pump used), case (A).

1st Membrane Surface	m^2^	200,000 (A_1_)		400,000 (A_2_)			500,000 (A_3_)		
1st compressor pressure	Bar	8	10	6	8	10	6	8	10
Case abbreviation	-	A_12_	A_13_	A_21_	A_22_	A_23_	A_31_	A_32_	A_33_
CO_2_ efficiency	%	73.14	87.43	75.85	94.16	99.23	84.14	97.82	99.9
CO_2_ purity	%	56.67	54.63	49.03	45.62	40.49	46.05	41.06	35.41
Power consumption	MW	133	156	126	159	185	135	169	198
CO_2_ captured/Membrane area	kmol/m^2^·h	0.019	0.023	0.010	0.012	0.013	0.008	0.010	0.010

**Table 10 membranes-12-00904-t010:** The evaluation of the CFPP system with a single-stage of membrane technology (no vacuum pump used).

Parameters	Power Plant	A_12_	A_13_	A_21_	A_22_	A_23_	A_31_	A_32_	A_33_
Fuel feedstock, (t/h)	92.22	92.22	92.22	92.22	92.22	92.22	92.22	92.22	92.22
CO_2_ capture efficiency, (%)	n.a.	73.14	87.43	75.85	94.16	99.23	84.14	97.82	99.90
Net power generated, (MW)	330	202.5	179.3	210	177	150.5	201	166	137.5
Net power plant efficiency, (%)	45.78	31.72	28.51	32.74	28.18	24.51	31.46	26.67	22.71
Capital costs per net electrical capacity, (€/kWh)	2754	5347	6037	5265	6243	7340	5562	67,189	8112
CO_2_ emission factor, (kg/MWh)	741.15	324.43	171.42	281.51	80.73	12.51	193.39	32.11	1.78
CO_2_ captured, (kg/MWh)	n.a.	883.41	1192	884.15	1302	1613	1026	1441	1777
Power consumption of membrane plant, (kWe)	n.a.	132,990	156,140	125,660	158,550	184,980	134,900	169,430	197,950
Membrane power consumption, (kWh/tCO_2_)	n.a.	743.43	730.18	677.36	688.45	762.18	655.52	708.17	810.15
LCOE_tax, (€/kWh)	0.0756	0.1372	0.1404	0.1313	0.1370	0.1555	0.1306	0.1431	0.1708
SPECCA_m_, (MJth/kg)	n.a.	3.94	3.99	3.54	3.77	4.23	3.49	3.91	4.50
SEPCCA_s_, (MJel/kg)	n.a.	2.14	2.14	1.90	2.00	2.28	1.85	2.08	2.45
CO_2_ avoided cost (€/t)	n.a.	147.92	113.79	121.29	92.95	109.64	100.34	95.14	128.82
CO_2_ captured cost (€/t)	n.a.	69.77	54.37	63.05	47.16	49.54	53.57	46.82	53.61

**Table 11 membranes-12-00904-t011:** The economical assessment of the CFPP system with a single-stage of membrane technology (no vacuum pump used).

Parameters	Units	A_12_	A_13_	A_21_	A_22_	A_23_	A_31_	A_32_	A_33_
NPV	[M€]	813	963	922	1087	985	1035	1073	880
VNA	[M€]	753	892	853	1006	912	958	994	815
IRR	[%]	0.16	0.17	0.17	0.18	0.17	0.18	0.18	0.16
UPP	[year]	7.60	7.18	7.34	6.93	7.18	7.08	6.99	7.48
DPP	[year]	10.16	9.34	9.65	8.87	9.33	9.14	8.97	9.91
PI	[-]	1.75	1.89	1.83	1.98	1.89	1.93	1.96	1.79

**Table 12 membranes-12-00904-t012:** The specific parameters were chosen for the technical and economical assessment of a single- stage of membrane technology (with a vacuum pump), case (B).

1st Membrane Surface	m^2^	200,000 (B_1_)			400,000 (B_2_)			500,000 (B_3_)		
1st compressor pressure	Bar	6	8	10	6	8	10	6	8	10
Case abbreviation	-	B_11_	B_12_	B_13_	B_21_	B_22_	B_23_	B_31_	B_32_	B_33_
CO_2_ efficiency	%	88.27	96.17	98.93	98.78	99.9	100	99.70	99.99	100
CO_2_ purity	%	55.45	56.67	54.63	51.88	44.35	38.6	46.48	38.92	33.47
Power consumption	MW	142	171	195	173	209	242	187	228	265
CO_2_ captured/Membrane area	kmol/m^2^·h	0.023	0.025	0.026	0.013	0.013	0.013	0.010	0.010	0.010

**Table 13 membranes-12-00904-t013:** The evaluation and economical assessment of the CFPP system with a single-stage of membrane technology (vacuum pump used).

Parameters	Power Plant	B_11_	B_12_	B_13_	B_21_	B_22_	B_23_	B_31_	B_32_	B_33_
Fuel feedstock, (t/h)	92.22	92.22	92.22	92.22	92.22	92.22	92.22	92.22	92.22	92.22
CO_2_ capture efficiency, (%)	n.a.	88.27	96.17	98.93	98.78	99.90	100	99.70	99.99	100
Net power generated, (MW)	330	194	164.5	140.5	162	126	93.5	148	107.7	70
Net power plant efficiency, (%)	45.78	30.49	26.46	23.13	26.14	21.16	16.60	24.21	18.57	13.35
Capital costs per net electrical capacity, (€/kWh)	2754	5592	6581	7705	6809	8744	11,819	7521	10,360	15,932
CO_2_ emission factor, (kg/MWh)	741.15	148.19	56.94	18.62	18.39	1.94	0.00	4.95	0.23	0.00
CO_2_ captured, (kg/MWh)	n.a.	1115	1429.7	1722	1489	1934	2617	1644	2271	2493
Power consumption of membrane plant, (kWe)	n.a.	141,890	170,960	194,970	173,240	209,150	242,020	187,150	227,790	265,460
Membrane power consumption, (kWh/tCO_2_)	n.a.	657.22	726.82	805.77	717.06	855.98	989.52	767.48	931.43	1085.4
LCOE_tax, (€/kWh)	0.0756	0.13	0.1434	0.1650	0.1444	0.1852	0.2518	0.1583	0.22	0.34
SPECCA_m_, (MJth/kg)	n.a.	3.54	4.02	4.47	3.97	4.73	5.28	4.26	5.07	5.47
SEPCCA_s_, (MJel/kg)	n.a.	1.87	2.15	2.43	2.11	2.62	3.10	2.30	2.89	3.44
CO_2_ avoided cost (€/t)	n.a.	89.91	99.12	123.75	95.21	148.21	237.70	112.35	194.04	356.69
CO_2_ captured cost (€/t)	n.a.	47.81	47.44	51.93	46.21	56.65	67.32	50.32	63.31	75.69

**Table 14 membranes-12-00904-t014:** The economic assessment of the CFPP system with a single-stage of membrane technology (vacuum pump used).

Parameters	Units	B_11_	B_12_	B_13_	B_21_	B_22_	B_23_	B_31_	B_32_	B_33_
NPV	[M€]	1106	1050	912	1074	794	516	968	627	306
VNA	[M€]	1024	972	844	994	736	478	896	581	283
IRR	[%]	0.19	0.18	0.17	0.18	0.16	0.13	0.17	0.14	0.11
UPP	[year]	6.84	6.97	7.32	6.96	7.71	8.72	7.25	8.30	9.76
DPP	[year]	8.70	8.93	9.60	8.92	10.38	12.64	9.46	11.65	15.5
PI	[-]	2.02	1.97	1.84	1.97	1.72	1.47	1.87	1.56	1.27

**Table 15 membranes-12-00904-t015:** The specific parameters were chosen for the technical and economical assessment of 2-stages of membrane technology, case (C).

1st Membrane Surface	m^2^	200,000 (C_1_)			400,000 (C_2_)			600,000 (C_3_)		
1st compressor pressure	Bar	6	8	10	6	8	10	6	8	10
Case abbreviation	-	C_11_	C_12_	C_13_	C_21_	C_22_	C_23_	C_31_	C_32_	C_33_
CO_2_ efficiency	%	43.21	65.78	80.57	64.52	86.08	95.56	78.00	94.85	99.26
CO_2_ purity	%	91.34	94.74	95.99	94.61	96.36	96.89	95.81	96.85	97.07
Power consumption	MW	102	133	160	121	162	198	140	189	233
CO_2_ captured/Membrane area	kmol/m^2^·h	0.011	0.017	0.021	0.009	0.011	0.012	0.007	0.008	0.009

**Table 16 membranes-12-00904-t016:** The evaluation and economical assessment of the CFPP system with 2-stages of membrane technology.

Parameters	Power Plant	C_11_	C_12_	C_13_	C_21_	C_22_	C_23_	C_31_	C_32_	C_33_
Fuel feedstock, (t/h)	92.22	92.22	92.22	92.22	92.22	92.22	92.22	92.22	92.22	92.22
CO_2_ capture efficiency, (%)	n.a.	43.21	65.78	80.57	64.52	86.08	95.56	78.00	94.85	99.26
Net power generated, (MW)	330	233	203	176	214	174	138	196	147	102
Net power plant efficiency, (%)	45.78	36.00	31.74	28.00	33.36	27.75	22.71	30.81	23.96	17.79
Capital costs per net electrical capacity, (€/kWh)	2754	4659	5365	6188	5176	6380	8066	5774	7717	11,085
CO_2_ emission factor, (kg/MWh)	741.15	595.25	413.08	270.53	404.98	195.85	78.97	274.67	85.94	17.74
CO_2_ captured, (kg/MWh)	n.a.	452.91	794.05	1,121.79	736.46	1 211	1 700	973.83	1 583	2 379
Power consumption of membrane plant, (MWe)	n.a.	102	133	160	121	162	198	140	189	233
Membrane power consumption, (kWh/tCO_2_)	n.a.	966.46	825.86	811.02	768.10	767.80	847.07	731.67	814.36	961.61
LCOE_tax, (€/kWh)	0.0756	0.14	0.14	0.15	0.14	0.15	0.18	0.14	0.17	0.24
SPECCA_m_, (MJth/kg)	n.a.	4.71	4.38	4.45	3.98	4.22	4.70	3.92	4.52	5.20
SEPCCA_s_, (MJel/kg)	n.a.	2.82	2.43	2.43	2.19	2.28	2.58	2.12	2.46	3.00
CO_2_ avoided cost (€/t)	n.a.	462.77	209.65	160.77	188.22	134.10	151.91	139.99	141.19	220.86
CO_2_ captured cost (€/t)	n.a.	149.07	86.62	67.45	85.91	60.38	59.18	67.06	58.44	67.15

**Table 17 membranes-12-00904-t017:** The economic assessment of the CFPP system with 2-stages of membrane technology.

Parameters	Units	C_11_	C _12_	C _13_	C _21_	C _22_	C _23_	C _31_	C _32_	C _33_
NPV	[M€]	343	630	761	679	859	780	831	820	547
IRR	[%]	0.12	0.14	0.15	0.15	0.16	0.16	0.16	0.16	0.13
UPP	[year]	9.51	8.22	7.78	8.09	7.53	7.76	7.66	7.69	8.64
DPP	[year]	14.76	11.49	10.52	11.20	10.00	10.49	10.27	10.34	12.45
PI	[-]	1.32	1.58	1.70	1.61	1.77	1.70	1.73	1.73	1.48

**Table 18 membranes-12-00904-t018:** A comparison 1- and 2-stages of membrane regarding different parameters.

Parameters	Units	1-Stage (without Vacuum)	1-Stage (with Vacuum)	2-Stages
CO_2_ capture efficiency	%	94.16	96.17	94.85
CO_2_ purity	%	45.62	56.67	96.85
Power consumption	MW	159	171	189
LCOE_tax	€/kWh	0.1370	0.1434	0.17
CO_2_ avoided cost	€/t	92.95	99.12	141.19
CO_2_ captured cost	€/t	47.16	47.44	58.44
NPV	M€	1087	1050	820

**Table 19 membranes-12-00904-t019:** The comparison of the current paper and different papers regarding technical and economical parameters.

Parameters	Current Study	Study _1_	Study _2_	Study _3_	Study _4_
Fuel feedstock, [t/h]	92.22	n.a.	n.a.	n.a.	n.a.
CO_2_ capture efficiency, [%]	94.85	90.01	67	90	90
CO_2_ purity, [%]	96.85	95.67	88	95	95
CO_2_ permeance, [GPU]	1000	740	270	100	1000
CO_2_/N_2_ selectivity	50	135	34	43	50
1st, 2nd Compressor pressure, [bar]	8, 4	2.5	5	1	n.a.
1st, 2nd vacuum pump pressure, [bar]	No utilization	0.25	n.a.	n.a.	n.a.
Flue gas flow, [kmol/h]	40,320	9580	6690	n.a.	500 *
Net power plant efficiency, [%]	23.96	35.5	n.a.	n.a.	n.a.
Capital costs per net electrical capacity, [€/kWh]	7717	n.a.	n.a.	n.a.	n.a.
CO_2_ emission factor, [kg/MWh]	85.94	n.a.	n.a.	n.a.	n.a.
Power consumption of membrane plant, [MWe]	189	165	137	n.a.	145
LCOE_tax, [€/kWh]	0.17	n.a.	n.a.	n.a.	n.a.
SPECCA_m_, [MJth/kg]	4.52	n.a.	n.a.	n.a.	n.a.
SEPCCA_s_, [MJel/kg]	2.46	1.09	n.a.	n.a.	n.a.
CO_2_ avoided cost [€/t]	141.19	47.40	197	n.a.	n.a.
CO_2_ captured cost [€/t]	58.44	47.11	n.a.	72	39

* m^3^/s.

## Data Availability

Not applicable.

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
