# Peer review of "Critical Assessment of Membrane Technology Integration in a Coal-Fired Power Plant"

_membranes, 2022, doi:10.3390/membranes12090904_

Round 1
Reviewer 1 Report
This manuscript presents a critical assessment of membrane technology integration in a coal-fired power plant. The author studies 3 different scenarios of CO2 capture using membranes and presents the process efficiency along with their economic implications.
· In its current version, there are many grammatical errors in the manuscript starting from abstract to conclusion sections, must be corrected before to be published in this journal.
· Expand the nomenclature as many terms are not listed in it, especially from process flow diagrams part. Make abbreviations separately, it should not be the part of paragraphs.
· The author talks about the materials for membranes synthesis in the section 2, it seems that there is no correlation of it with the CO2 capture efficiency and economic analysis presented in the last section.
· As there are many recent similar studies present, modeled using CHEMCAD, is it possible to provide a clear picture and compare the findings of this study with some of them with respect to membranes incorporation for CO2 capture in coal power plant studies.
· To reduce the cost, is it possible to perform modeling of hybrid membranes processes by selecting the optimized surface area (SA) of membrane for CO2 capture using CHEMMCAD?
· Is it possible to include and model the effects of CO2 production on the environment and its consequences in terms of economic losses?
Author Response
Dear Reviewer,
First, we would like to thank you for your worthy comments, which will significantly improve the article.
The introduction sections have been revised and improved. Where several phrases regarding CO2 capture process and its performance compared with the classical amine absorption process have been added to the section.
The references section has been revised and modified depending on the edited and additional information that added to the paper.
The method section has been revised and edited by adding more phrases regarding CO2 purity and 70 bar compressor, and the title of the section (page 9) has been changed into ‘Membrane technology for CO2 capture’
The author has revised and enhanced the description of methods and results. Where the variations parameters in table 7 has been reordered and titled as ‘Variations of different membrane parameters simulated’ to clarify the different parameters simulated. The result description has been modified as well by presenting the effect of membrane surface on CO2 purity in page 14 and editing specific figures and tables descriptions, such as figure 20 in page 20 and table 12 in page 26.
The conclusion section has been revised and enhanced, the effect of increasing 1st compressor pressure on LCOE has been added.
Note/ The inquiries of the reviewer have been marked with a green highlight and the answers with a yellow one.
Reviewer 1
Q1- In its current version, there are many grammatical errors in the manuscript starting from abstract to conclusion sections, must be corrected before to be published in this journal.
Answer- The grammar and punctuation have been revised and improved.
Q2- Expand the nomenclature as many terms are not listed in it, especially from process flow diagrams part. Make abbreviations separately, it should not be the part of paragraphs.
Answer- The nomenclatures of the process flow diagram have been added to the nomenclature section on page 33.
Q3- As there are many recent similar studies present, modeled using CHEMCAD, is it possible to provide a clear picture and compare the findings of this study with some of them with respect to membranes incorporation for CO2 capture in coal power plant studies.
Answer- A comparison between the results of the present paper and other scientific papers has been added to table (18) on page 31.
Q4- To reduce the cost, is it possible to perform modeling of hybrid membranes processes by selecting the optimized surface area (SA) of membrane for CO2 capture using CHEMMCAD?
Answer- In our simulation, we take into consideration different parameters (e.g. pressure, CO2 purity, etc.) impacting the cost besides the membrane surface such as 1st compressor pressure.
The current critical assessment simulation is the first research in our laboratory related to CO2 capture hybrid processes with membrane technology. However, once we select the right membrane for CO2 separation from flue gas, we will be able to optimize the membrane surface under conditions of efficiency and purity.
We are conscious that decreasing the membrane surface leads to reducing the cost but at the same time influences the capture efficiency and purity of CO2, where the purpose of the paper is to obtain at least 90% CO2 capture efficiency and at least 95% of CO2 purity as improved and demonstrated on page 9.
Q5- Is it possible to include and model the effects of CO2 production on the environment and its consequences in terms of economic losses?
Answer- The aim of this research is to evaluate performances (technically and economically) of the membrane integration into a coal-fired power plant. We understand that to model the effects of CO2 production on the environment and its consequences is another subject.
Reviewer 2
Q1- Figures 5 - 7 may have incorrect symbols for the membrane separation module, compressor, and vacuum pump.
Answer- Figures 5-7 on page 10 have been revised and improved by adding a (VP) symbol to vacuum pump equipment and 70 bar representing the last compressor before transportation.
Q2- Authors stated the compressor and vacuum pump efficiencies of 90%. Please offer the related references. Generally, the efficiency reaches 80% to 85% in the reported literature.
Answer- The current paper focused on using membrane technology integrated into CFPP in the coming years (at least 5 years), in that time, regarding the improvements of pieces of equipment and performance, we believe the efficiency of equipment (such as compressors and vacuum pumps) could be enhanced. However, the difference between using a compressor or pump efficiency of 90% in-stead of 85% leads to 6% decrease in the energy consumption at the CO2 capture efficiency of 90%. We mentioned all these aspects at the page 28, the third paragraph.
Q3- The primary objective of this work is to assess the membrane process, but greater emphasis is devoted to membrane materials.
Answer- Our objective is to check if these membrane materials are developed in CO2 hybrid projects and suitable for coal-fired power plants. We emphasizes the materials parametrizations in section 2 because we tried to find different data like permeability and selectivity related to the materials; there are parameters that we have already used in our modeling in CHEMCAD.
Q4- The simulation employs the membrane with a fixed CO2 permeance of 1000 GPU and a CO2/N2 selectivity of 50. However, the authors seem not considering the pressure-dependent variation in the membrane performance.
Answer- We are willing to examine the durability and the presence of impurities that impact the performance of the material but currently, we don’t have an equation that figures out this dependency, so in our paper, we considered that the performance of the material is not affected by the pressure variation during the process
Q5- Please provide an explanation and a source for the membrane material.
There are several technical and economic studies of single-stage and two-stage simulations of membrane processes in the scientific literature.
What distinguishes this paper from others reported in literature?
Answer- On page 10, the membrane material has been revised and references have been edited and added, and the following sentences have been added to the paper as well:
‘Generally, the lifetime of the membrane is 5 years, where after this period the membrane performance will decrease and the material must be replaced.’
The added value of the article: The article takes into account the study of the technical and economic indicators regarding the specific parameters simulated (e.g. Compressors, vacuum pumps, and membrane surface area) of a membrane process integrated into a coal-fired
Q6- Figure 4 contains abbreviations, such as CFBC, HPST. Please provide the explanation for them.
Answer- The abbreviations of the figure (4) such as CFBC, HPST, …etc. have been added to the nomenclature section on page 33.
Q7- Please include the required membrane area per ton of CO2 capture in Table 11
Answer- In tables (8, 11, 14) which the specific parameters of 1-stage of the membrane (with and without a vacuum pump used) and 2-stages are demonstrated, the CO2 captured/Membrane area parameter has been calculated and added for each table to present a clear comparison of every case, the tables start from page 23.
Reviewer 3
Q1- The authors showed that increasing the membrane area when using 1-stage membrane process can improve the CO2 capture efficiency and purity, but it will increase the investment cost. It would be better to compare the investment cost and other economic cost such as energy consumption of 1-stage and 2-satge membrane processes with the optimal CO2 capture efficiency and purity
Answer- A comparison between 1-stage and 2-stages of the membrane considering the preferable capture efficiency and purity of CO2 has been added to table 17 on page 30. The other compared parameters are power consumption, LCOE, CO2 avoided and captured cost, and net present value (NPV).
Q2- How about is the stability of the five types of polymer membranes mentioned in this work during the CO2 capture process, and how to enhance the membrane stability of 1-stage or 2-stages?
Answer- In our CO2 hybrid project we study the durability of the membrane material integrated into a coal-fired power plant in order to see how the material behavior is affected in the presence of dust, sulfur dioxide, nitrogen dioxide, ...etc. But this is not the purpose of the current paper, in this paper, we discuss only one membrane material considering the parameters used for the simulation.
Q3- Why does the membrane area affect the capture efficiency and purity of CO2? What is the purpose of using a 70 bar compressor with a cooling center during level 4?
Answer-The membrane surface area has a significant effect on the capture efficiency and purity of CO2 where increasing the surface helps to raise the CO2 contents flow passed through the membrane module, which increases the efficiency of CO2 capture as demonstrated in figure (15). In terms of CO2 purity, if the membrane surface area is higher than the optimal one, the other molecules than CO2 (e.g. N2) passed via the membrane which decreases the CO2 purity, as described in figure (13).
As the CO2 transportation part revised and improved on page 10, when the coal-fired power plant is analyzed by the CO2 capture process the CO2 contents captured must be prepared at high pressure for transportation goals, where the CO2 captured stream must be compressed. However, different other high pressures could be used but we consider the pressure is 70 bar.
The Cooling center is used to mitigate the high temperature resulting from the compressor, demonstrated clearly in section 3, page 10.

Reviewer 2 Report
This manuscript reported a critical assessment of membrane technology integration in a 2 coal-fired power plant. Generally, the investigation content is sufficient and the research work is also meaningful for gas separation membrane application. Thus, the manuscript could be accepted after considering the following comments.
1. Figures 5 - 7 may have incorrect symbols for the membrane separation module, compressor, and vacuum pump.
2. Authors stated the compressor and vacuum pump efficiencies of 90%. Please offer the related references. Generally, the efficiency reaches 80% to 85% in the reported literature.
3. The primary objective of this work is to assess the membrane process, but greater emphasis is devoted to membrane materials.
The simulation employs the membrane with a fixed CO2 permeance of 1000 GPU and a CO2/N2 selectivity of 50. However, the authors seem not considering the pressure-dependent variation in the membrane performance.
4. Please provide an explanation and a source for the membrane material.
There are several technical and economic studies of single-stage and two-stage simulations of membrane processes in the scientific literature.
What distinguishes this paper from others reported in literature?
5. Figure 4 contains abbreviations, such as CFBC, HPST. Please provide the explanation for them.
6. Please include the required membrane area per ton of CO2 capture in Table 11.
Author Response

(The authors gave the same response as above.)

Reviewer 3 Report
By analyzing the technical and economic integration of the spiral wound membrane in the coal-fired power plant with an installed capacity of 330MW, and using the energy simulation process to evaluate and compare the 1-stage and 2-stage of membrane performance, it is indicated that using vacuum pump and 2-stage membrane process shows higher CO2 capture efficiency and purity. Then, the authors used three different schemes with a large number of parameters to achieve the targets of 90% capture efficiency and at least 95% purity for CO2 capture. Following a serial of characterizations, it was demonstrated that using vacuum pump and lower membrane area will increase CO2 capture efficiency and purity. The article is clear and comprehensive, but there are some revisions need to be addressed.
1、The authors showed that increasing the membrane area when using 1-stage membrane process can improve the CO2 capture efficiency and purity, but it will increase the investment cost. It would be better to compare the investment cost and other economic cost such as energy consumption of 1-stage and 2-satge membrane processes with the optimal CO2 capture efficiency and purity.
2、How about is the stability of the five types of polymer membranes mentioned in this work during the CO2 capture process, and how to enhance the membrane stability of 1-stage or 2-stages?
3、Why does the membrane area affect the capture efficiency and purity of CO2? What is the purpose of using a 70 bar compressor with a cooling center during level 4?
Author Response

(The authors gave the same response as above.)

Round 2
Reviewer 1 Report
The manuscript is now revised addressing all the comments. It may be accepted.